# Auditing Predictive Models for Intersectional Biases

**Kate S. Boxer**                                                                     *kb145@nyu.edu*
*Department of Computer Science*
*New York University*

**Edward McFowland III**                                                       *emcfowland@hbs.edu*
*Technology and Operations Management*
*Harvard Business School*

**Daniel B. Neill**                                                             *daniel.neill@nyu.edu*
*Department of Computer Science*
*New York University*

**Reviewed on OpenReview:** *https://openreview.net/forum?id=1JTnlHMSmO*

## Abstract

Predictive models that satisfy group fairness criteria in aggregate for members of a protected class, but do not guarantee subgroup fairness, could produce biased predictions for individuals at the intersection of two or more protected classes. To address this risk, we propose Conditional Bias Scan (CBS), an auditing framework for detecting intersectional biases in the outputs of classification models that may lead to disparate impact. CBS aims to identify the subgroup with the most significant bias against the protected class, compared to the equivalent subgroup in the non-protected class. The framework can audit for predictive biases using common group fairness definitions that can be represented as conditional independence statements (separation and sufficiency) for both probabilistic and binarized predictions. We show through empirical evaluations that this methodology has substantially higher bias detection power compared to similar methods that audit for subgroup fairness. We then use this approach to detect statistically significant intersectional biases in the predictions of the COMPAS pre-trial risk assessment tool and a model trained on the German Credit data.

## 1 Introduction

Predictive models are increasingly used to assist in high-stakes decisions and, therefore, can have significant impacts on individuals' lives and livelihoods. However, recent studies have revealed numerous models whose predictions contain biases, in the form of group fairness violations, against disadvantaged and marginalized groups (Angwin et al., 2016a; Obermeyer et al., 2019). When auditing a predictive model for bias, typical group fairness definitions (Mitchell et al., 2021) rely on univariate measurements of the difference between the distributions of predictions or outcomes for individuals in a "protected class", typically defined by a sensitive attribute such as race or gender, as compared to those in the non-protected class. Since these approaches only detect biases for a predetermined subpopulation at an aggregate level, e.g., a bias against Black individuals, they may fail to detect biases that adversely affect a subset of individuals in a protected class, e.g., Black females.

While it is possible to define a specific multidimensional subgroup and then audit a classifier for biases impacting that subgroup, this approach relies on defining a fixed subgroup and therefore is limited to static queries. From a computational perspective, this approach does not scale to the exponential number of multidimensional subgroups; from a statistical perspective, it fails to account for the risk of identifying many false positives if many subgroups are tested for bias. To detect whether there are *any* subgroups within a given protected class that are adversely impacted by predictive biases, we need a way to *search* over

the combinatorial number of subgroups of the protected class, and to account for this search process when deciding which observed biases are statistically significant, rather than separately auditing each subgroup.

To address these issues, we present Conditional Bias Scan (CBS), a novel and flexible framework for bias detection. Given a classifier's probabilistic *predictions* or binarized *recommendations* based on those predictions, CBS discovers systematic biases impacting any *subgroups* of a predefined subpopulation of interest (the *protected class*). More precisely, CBS aims to discover subgroups of the protected class for whom the classifier's predictions or recommendations systematically deviate from the corresponding subgroup of individuals who are not a part of the protected class. Subgroups are defined by a non-empty subset of attribute values for each observed attribute, excluding the *sensitive attribute* which determines whether or not individuals belong to the protected class. We constrain subgroups to categorical representations of the non-sensitive attributes for explainability purposes. This requires discretization of continuous variables prior to the scanning step of our methodology, as discussed in detail below. To provide a concrete example, the binary sensitive attribute could be defined by whether or not an individual is female, and CBS would search over all other attributes (e.g., race/ethnicity and income) to identify subgroups with a gender disparity between female and non-female individuals. If CBS returns a subgroup consisting of Black or Hispanic individuals whose yearly income is $0-$9,999 or $10,000-$25,000, this would indicate that the most significant gender disparity identified by the scan affected individuals with this combination of race/ethnicity and low income. We provide formal notation for a sensitive attribute and subgroup at the start of Section 2.

The detected subgroups can represent both *intersectional* and *contextual* biases. In this paper, as in much of the algorithmic fairness literature, we refer to *intersectional* biases when individuals of a subgroup are members of two or more protected classes. We refer to *contextual* biases for forms of subgroup biases that may only be present for certain decision situations (Runyan, 2018). In Section 5.4, we discuss the distinct, yet related topic of intersectionality theory (Crenshaw, 1991a;b; Collins, 2008) in sociology, which describes how individuals' different social positions and identities interact to influence their social and material conditions. In particular, an individual at the intersection of several marginalized groups may be impacted by multiple historical and continuing systems of power and oppression including structural racism, sexism, and class stratification.

The contributions of our research include:

- A methodological framework that can flexibly accommodate multiple group fairness definitions and can effectively detect intersectional and contextual biases.

- A computationally efficient detection algorithm to audit classifiers for fairness violations in the exponentially many subgroups of a given subpopulation.

- Detailed empirical evaluations demonstrating substantially improved bias detection accuracy as compared to similar methods that audit for subgroup fairness, along with real-world case studies on COMPAS and German Credit datasets.

Given the overarching objective of understanding the full scope of predictive biases that a model produces for *all* the sensitive subgroups of a given target population, there is a need for expanded measurements of predictive bias and improved methods for searching for these biases within all sensitive subgroups that could be adversely affected by predictive bias. Without auditing tools that can accurately identify these biases, any predictive bias definition will be limited to evaluating a small, static set of subgroups, and there will presumably be some form of intersectional or contextual bias that goes undetected. Therefore, CBS is an important step towards understanding the full scope of biases that a predictive model might produce.

## 2 Methods

We define the dataset $D = (A, X, Y, P, P_{bin}) = \{(A_i, X_i, Y_i, P_i, P_{i,bin})\}_{i=1}^n$, for $n$ individuals indexed as $i = 1 \ldots n$. The sensitive attribute, $A_i$, is a binary variable representing whether individual $i$ belongs to the protected class. $X_i = (X_i^1, \ldots, X_i^m)$ are other covariates for individual $i$, excluding $A_i$. $Y_i$ is individual $i$'s observed binary outcome, $P_i \in [0, 1]$ is the classifier's probabilistic prediction of individual $i$'s outcome, and

Table 1: Table of scan types for CBS for different group fairness definitions. Scans that correspond to well-known fairness definitions (e.g., false positive error rate balance) are noted in the table. Shading is used to distinguish between the four main categories of scans (separation for predictions, separation for recommendations, sufficiency for predictions, and sufficiency for recommendations). The notation $\perp$ refers to conditional independence from membership in the protected class ($A$).

| | | Predictions ($P \in [0,1]$) | Recommendations ($P_{bin} \in \{0,1\}$) | |
| --- | --- | --- | --- | --- |
| | | | $P_{bin} = 1$ | $P_{bin} = 0$ |
| Separation | $Y = 1$ | $\mathbb{E}[P \mid Y = 1, X]\perp A$ *Balance for Positive Class* | $\Pr(P_{bin} = 1 \mid Y = 1, X)\perp A$ *True Positive Rate* | $\Pr(P_{bin} = 0 \mid Y = 1, X)\perp A$ *False Negative Rate* |
| | $Y = 0$ | $\mathbb{E}[P \mid Y = 0, X]\perp A$ *Balance for Negative Class* | $\Pr(P_{bin} = 1 \mid Y = 0, X)\perp A$ *False Positive Rate* | $\Pr(P_{bin} = 0 \mid Y = 0, X)\perp A$ *True Negative Rate* |
| Sufficiency | $Y = 1$ | $\Pr(Y = 1 \mid P, X)\perp A$ | $\Pr(Y = 1 \mid P_{bin} = 1, X)\perp A$ *Positive Predictive Value* | $\Pr(Y = 1 \mid P_{bin} = 0, X)\perp A$ *False Omission Rate* |
| | $Y = 0$ | $\Pr(Y = 0 \mid P, X)\perp A$ | $\Pr(Y = 0 \mid P_{bin} = 1, X)\perp A$ *False Discovery Rate* | $\Pr(Y = 0 \mid P_{bin} = 0, X)\perp A$ *Negative Predictive Value* |

$P_{i,bin} \in \{0,1\}$ is the binary recommendation[1] corresponding to $P_i$. For example, it is common to define $P_{i,bin} = \mathbf{1}\{P_i \geq 0.5\}$.

Given these data, the CBS framework searches for subgroups of the protected class, defined by a subset of values for each covariate $X^1, \ldots, X^m$, for whom some *group fairness definition* (contained in Table 1) is violated with respect to $A$. Therefore, CBS returns a subgroup $S$ represented as the Cartesian product, $S = S^1 \times S^2 \times \ldots \times S^m$, where $S^j \subseteq X^j$ for $j = 1, \ldots, m$. Each fairness definition in Table 1 is in the form of a conditional independence relationship between an individual's membership in the protected class, $A_i$, and their value of an *event variable*, $I_i$, conditioned on their value of a *conditional variable*, $C_i$, and their covariates, $X_i$: $\mathbb{E}[I_i \mid C_i, X_i]\perp A_i$. We define the null hypothesis, $H_0$, that $I \perp A \mid (C, X)$, and use CBS to search for subgroups with statistically significant violations of this conditional independence relationship, correctly adjusting for multiple hypothesis testing, allowing us to reject $H_0$ in favor of the alternative hypothesis $H_1$ that $I \not\perp A \mid (C, X)$.

The CBS framework has four sequential steps:

(1) Given a fairness definition, CBS chooses $I \in \{Y, P, P_{bin}\}$ and $C \in \{Y, P, P_{bin}\}$. **Section 2.1** maps different group fairness criteria to particular choices of event variable $I$ and conditional variable $C$.

(2) CBS estimates the expected value of $I_i$ for each individual in the protected class under the null hypothesis $H_0$ that $I$ and $A$ are conditionally independent given $C$ and $X$. These expectations are denoted as $\hat{I}_i$, and **Section 2.2** describes how to estimate $\hat{I}$.

(3) CBS uses a novel *multidimensional subset scan* to search for subgroups, $S$, where, for $i \in S$, the observed $I_i$ deviates systematically from its expectation $\hat{I}_i$ in the direction of interest. This step to *detect $S^*$* is described in **Section 2.3**.

(4) The final step to *evaluate statistical significance* of the detected subgroup $S^*$, in **Section 2.4**, uses permutation testing to adjust for multiple hypothesis testing and determine if $S^*$'s deviation between protected and non-protected class is statistically significant.

---

[1] $P_{i,bin}$ is sometimes referred to as a binary prediction. We use the term "recommendation" to distinguish $P_{i,bin}$ from the probabilistic prediction $P_i$.

Note that, for Step (2), covariates $X$ can be represented as continuous or categorical variables; prior to Step (3), $X$ must be discretized and represented as categorical variables so that interpretable subgroups can be produced by the scan.

## 2.1 Define $(I, C)$: *Overview of Scan Types*

Many of the group fairness criteria proposed in the fairness literature fall into two categories of statistical fairness called sufficiency and separation. *Sufficiency* is focused on equivalency in the rate of an outcome (for comparable individuals with the same prediction or recommendation) regardless of protected class membership ($Y \perp A \mid P, X$), whereas *separation* is focused on equivalency of the expected prediction or recommendation (for comparable individuals with the same outcome) regardless of protected class membership ($P \perp A \mid Y, X$).

The choice between separation and sufficiency determines whether outcome $Y$ is the event variable of interest $I$ or the conditional variable $C$, where bias is defined as $\mathbb{E}[I \mid C, X, A = 1] \neq \mathbb{E}[I \mid C, X, A = 0]$. The combination of fairness metric (sufficiency or separation) and prediction type (continuous prediction or binary recommendation) produces four classes of fairness scans, as defined by the chosen values for $I$ and $C$:

- For **separation** for **predictions**, we define $I = P$ and $C = Y$.

- For **separation** for **recommendations**, we define $I = P_{bin}$ and $C = Y$.

- For **sufficiency** for **predictions**, we define $I = Y$ and $C = P$.

- For **sufficiency** for **recommendations**, we define $I = Y$ and $C = P_{bin}$.

Depending on the particular bias of interest, we can also perform "value-conditional" scans by restricting the value of the conditional variable, $C$. For example, to scan for subgroups with increased false positive rate (FPR), we restrict the data to individuals with $Y = 0$, and perform a separation scan for recommendations (setting $I = P_{bin}$ and $C = Y$). All of the scan options for CBS are shown in Table 1. Each scan in Table 1 can detect bias in either direction, e.g., searching for subgroups with increased FPR (i.e., positive direction bias) or decreased FPR (i.e., negative direction bias).

## 2.2 Generate Expectations $\hat{I}$ of the Event Variable

Once we have defined the event variable $I$ and conditional variable $C$, as discussed in Section 2.1, we wish to detect fairness violations by assessing whether there exist subgroups of the protected class where $\mathbb{E}[I \mid C, X, A = 1]$ differs systematically from $\mathbb{E}[I \mid C, X, A = 0]$. For each individual $i$ in the protected class, $I_i \mid C_i, X_i, A_i = 1$ is observed but $I_i \mid C_i, X_i, A_i = 0$ is unobserved. Thus we calculate an estimate $\hat{I}_i = \mathbb{E}_{H_0}[I_i \mid C_i, X_i, A_i = 1]$, under the null hypothesis, $H_0$: $I \perp A \mid (C, X)$, and compare $\hat{I}_i$ to the observed $I_i$. Using the estimated $\hat{I}$ and observed $I$, we aim to determine which subgroups in the protected class have the largest deviations in $I$ as compared to what we would expect if there was no bias, $\hat{I}$. The method to generate $\hat{I}$ borrows from the literature on causal inference in observational settings, where propensity score reweighting is used to account for the selection of individuals into a "treatment" condition (here, membership in the protected class) given their observed covariates $X$.

The method to estimate $\hat{I}$ consists of the following steps:

Step 1: Train a predictive model using all the individuals in the data to estimate $\Pr(A = 1 \mid X)$.

Step 2: Use this model to produce the probabilities, $p_i^A = \Pr(A_i = 1 \mid X_i)$, and the corresponding propensity score weights, $w_i^A = \frac{p_i^A}{1 - p_i^A}$, for each individual $i$ in the non-protected class ($A_i = 0$). Intuitively, individuals in the non-protected class whose attributes $X_i$ are more similar to individuals in the protected class have higher weights $w_i^A$. This weighting scheme is used in the literature to produce causal effect estimates that can be interpreted as the average treatment effect on treated individuals (ATT) under typical causal inference assumptions of positivity and strong ignorability.

Step 3: Estimate $\mathbb{E}_{H_0}[I \mid C, X]$:

Case A (Binary event variable): **For all sufficiency scans** and **separation scan for recommendations**, where the event variable, $I$, is binary, we train a model using only data for individuals in the non-protected class ($A_i = 0$) to estimate $\mathbb{E}_{H_0}[I \mid C, X]$ by weighting each individual $i$ in the non-protected class by $w_i^A$. The trained model is used to estimate the expectations $\hat{I}_i = \mathbb{E}_{H_0}[I_i \mid C_i, X_i]$ for each individual in the protected class ($A_i = 1$) under the null hypothesis, $H_0$, of $I \perp A \mid (C, X)$.

Case B (Real-valued event variable): For the **separation scan for predictions**, where the event variable, $I$, is a real-valued variable, the probabilistic predictions $P$, rather than a binary event variable, we use a similar but modified process to estimate $\mathbb{E}_{H_0}[I \mid C, X]$, where $I = P$ and $C = Y$. For each individual $i$ in the non-protected class, we create two training records containing the same covariates, $X_i$, but different labels and associated weights:

  i. For the first record, we set the label, $I_{i_+}^{temp}$, equal to 1, and set the weight to $w_i^A P_i$.

  ii. For the second record, we set the label, $I_{i_-}^{temp}$, equal to 0, and set the weight to $w_i^A(1 - P_i)$.

Note, for a separation scan for predictions, $P_i$ is the *observed* prediction for individual $i$. We create a dataset that includes both records, described in i and ii above, for each individual in the non-protected class and their associated weights, and use this concatenated data set to train a model that estimates $\mathbb{E}_{H_0}[I^{temp} \mid C, X]$, by weighting each individual $i$ in the non-protected class by either $w_i^A P_i$ or $w_i^A(1 - P_i)$ as described above. This approach is consistent with other CBS variants and enforces the desired constraint $0 \leq \hat{I}_i \leq 1$, unlike alternative approaches such as using regression models to predict $\hat{I}$.

For value-conditional scans, such as the FPR, FDR, and balance for positive class scans shown in Table 1, CBS audits for biases in the subset of data where $C = z$, for $z \in \{0, 1\}$. Dataset $D$ is filtered before Step 3 to only include individuals where $C = z$. For example, for the value-conditional scan for FPR, we filter the data to only include individuals where $C = 0$ (or equivalently, $Y = 0$).

A probabilistic model can be used to estimate $\Pr(A = 1 \mid X)$ in Step 1, and a probabilistic model that allows for weighting of instances during training can be used to estimate $\mathbb{E}_{H_0}[I \mid C, X]$ in Step 3. For Sections 3 and 4, as well as Appendices B.3 and B.4, we use logistic regression to estimate $\Pr(A = 1 \mid X)$ and weighted logistic regression to estimate $\mathbb{E}_{H_0}[I \mid C, X]$. When estimating $\mathbb{E}_{H_0}[Y \mid P, X]$ (the realized expectation of $\mathbb{E}_{H_0}[I \mid C, X]$) for sufficiency scan for predictions, we transform the conditional variable, $P_i$, to its corresponding log-odds, $\log \frac{P_i}{1 - P_i}$, prior to training, since we expect $\log \frac{Y_i}{1 - Y_i}$ (the target of the logistic regression) to be approximately $\log \frac{P_i}{1 - P_i}$ for well-calibrated classifiers.

Accurate estimates of $\hat{I}$ are essential for CBS to accurately detect the subgroup in the protected class with the most deviation between the observed $I$ and estimated $\hat{I}$ under the null hypothesis of no bias. The method described above has the limitation of only producing accurate estimates of $\hat{I}$ when both the model for $\Pr(A = 1 \mid X)$ and $\mathbb{E}_{H_0}[I \mid C, X]$ are well-specified. Given the consistency of our findings for the COMPAS case study in Section 4 with other researchers' findings about COMPAS, as well as other checks we have performed to examine $\hat{I}$ (such as the calibration curve plots for both $\Pr(A = 1 \mid X)$ and $\mathbb{E}_{H_0}[I \mid C, X]$ for our COMPAS case study included in Appendix C.1.1), we believe that the method above suffices for COMPAS. However, we find that logistic regression is insufficient in estimating $\hat{I}$ for the German Credit Data, due to the smaller dataset size and highly-correlated predictors. Thus we use a more flexible model—a gradient boosting classifier with Platt scaling—in our German Credit Data experiments in Appendix C.2 to ensure that CBS predictions are well-calibrated when computing propensity scores and when estimating $\hat{I}$. In Appendix C.2.3, we include calibration curve plots for the models used to estimate $\Pr(A = 1 \mid X)$ and $\mathbb{E}_{H_0}[I \mid C, X]$ for our German Credit Data case study. We encourage others using CBS to be aware of this limitation, pay special consideration to estimates of $\hat{I}$, and if necessary, employ methods from the causal inference literature on doubly robust estimation (Imbens, 2004; Schuler & Rose, 2017) or methods from the computer science literature for model calibration when producing estimates of $\hat{I}$.

We note that both discrete-valued and continuous-valued covariates, $X$, can be used for estimating $\hat{I}$, for both the propensity model $\Pr(A = 1 \mid X)$ and the model of $\mathbb{E}_{H_0}[I \mid C, X]$. However, continuous-valued covariates,

Table 2: Null and alternative hypotheses, $H_0$ and $H_1(S)$, and corresponding log-likelihood ratio score functions, $F(S)$, used to measure a subgroup's degree of anomalousness (comparing the event variable $I$ to its expectation $\hat{I}$ under $H_0$) for all four variants of CBS. Shading is used to distinguish between Gaussian and Bernoulli scans and provide consistency with Table 1. Over-estimation (under-estimation) bias means that the expectations $\hat{I}_i$ are larger (smaller) than $I_i$. Note that the free parameters in $F(S)$ are $\mu$ and $q$ for Gaussian and Bernoulli scan respectively. Derivations of $F(S)$ can be found in Appendix A.2.

| Scan Types | | Hypotheses and Bias Constraints | | $F(S)$ |
|---|---|---|---|---|
| Separation | Predictions | $H_0:$ | $\Delta_i \sim N(0,\sigma),\ \forall i \in D_1$ | $\max_\mu \dfrac{2\mu\left(\sum_{i\in S}\Delta_i\right)-|S|\mu^2}{2\sigma^2}$ |
| | | $H_1(S):$ | $\Delta_i \sim N(\mu,\sigma)$ | |
| | | $where$ | $\Delta_i = \log\left(\frac{I_i}{1-I_i}\right) - \log\left(\frac{\hat{I}_i}{1-\hat{I}_i}\right)$ | |
| | | $Over\text{-}estimation:$ | $\mu < 0,\ \forall i \in S,$ and $\mu = 0,\ \forall i \notin S.$ | |
| | | $Under\text{-}estimation:$ | $\mu > 0,\ \forall i \in S,$ and $\mu = 0,\ \forall i \notin S.$ | $Gaussian\ Distribution$ |
| Sufficiency | Recommendations | $H_0:$ | $odds(I_i) = \frac{\hat{I}_i}{1-\hat{I}_i},\ \forall i \in D_1$ | $\max_q \sum_{i\in S}(I_i \log(q)$ |
| | Predictions | $H_1(S):$ | $odds(I_i) = q\frac{\hat{I}_i}{1-\hat{I}_i}$ | $-\log(q\hat{I}_i - \hat{I}_i + 1))$ |
| | Recommendations | $Over\text{-}estimation:$ | $q < 1,\ \forall i \in S,$ and $q = 1,\ \forall i \notin S.$ | |
| | | $Under\text{-}estimation:$ | $q > 1,\ \forall i \in S,$ and $q = 1,\ \forall i \notin S.$ | $Bernoulli\ Distribution$ |

$X$, must be discretized or removed prior to the scan step described below in Section 2.3, which requires that all scan dimensions, i.e. the covariates, $X$, are discrete-valued.

## 2.3 Detect the Most Significant Subgroup $S^*$

Given the observed event variables $I_i$ and the expectations $\hat{I}_i$ of the event variable under the null hypothesis $(I \perp A \mid C, X)$ for the protected class, where the procedure for calculating $\hat{I}$ is described in Section 2.2, we define a score function measuring *subgroup bias*, $F : S \to \mathbb{R}_{\geq 0}$, that can be efficiently optimized over exponentially many subgroups with the goal of identifying $S^* = \arg\max_S F(S)$.

To do so, we follow the literature on spatial and subset scan statistics (Kulldorff, 1997; Neill, 2012) by defining the general form of the various score functions, $F(S)$, for the CBS scans as a log-likelihood ratio (LLR) test statistic:

$$F(S) = \log\left(\frac{\Pr(D \mid H_1(S))}{\Pr(D \mid H_0)}\right) \tag{1}$$

Here the denominator represents the likelihood of seeing the observed values of event variable $I$ for subgroup $S$ of the protected class under the null hypothesis $H_0$ of no bias. The numerator represents the likelihood of seeing the observed values of $I$ for subgroup $S$ of the protected class under the alternative hypothesis $H_1(S)$, where the $I_i$ values are systematically increased or decreased as compared to $\hat{I}_i$.

For the alternative hypothesis, $H_1(S)$, to represent a deviation from $H_0$, $H_1$ contains a free parameter ($q$ or $\mu$) that is determined by maximum likelihood estimation. Under-estimation bias ($I_i > \hat{I}_i$) or over-estimation bias ($I_i < \hat{I}_i$) can be detected using different constraints for $q$ or $\mu$. **These constraints, as well as all**

**hypotheses ($H_0$ and $H_1$) and corresponding score functions, $F(S)$, for all scans are shown in Table 2.**

As shown in Table 2, when $I$ is a probabilistic prediction (i.e., for separation scan for predictions), the hypotheses are in the form of a difference of log-odds between $I$ and $\hat{I}$ sampled from a Gaussian distribution. Here, the free parameter $\mu$ in $H_1$ represents a mean shift ($\mu \neq 0$) of the Gaussian distribution. For all other scans, under $H_0$, each observed $I_i$ is assumed to be drawn from a Bernoulli distribution centered at the corresponding expectation $\hat{I}_i$. Under $H_1$, the free parameter $q$ represents a multiplicative increase or decrease ($q \neq 1$) of the odds of $I$ as compared to $\hat{I}$.

As shown in the rightmost column of Table 2, the various score functions all aggregate the deviations from $H_0$ for each instance in a subgroup, and thus the log-likelihood ratio score $F(S)$ scales linearly with subgroup size $|S|$ for a given amount of deviation between observed and expected $I$ values. This dependence on $|S|$ prevents the scan from assigning disproportionately high log-likelihood scores to subgroups with few instances that have large, chance deviations from the null hypothesis over favoring the true, larger subgroups of interest. For example, a subgroup with very few instances where there is a large deviation in, for example, the false positive rate between individuals in the protected class and those in the non-protected class, would not be favored over a larger subgroup with less pronounced deviation in its false positive rate because of this scaling effect that controls for subgroup cardinality.

As in Zhang & Neill (2016), a complexity penalty term can be added to the expressions for $F(S)$ shown in the rightmost column of Table 2. The complexity penalty is equal to a prespecified scalar times the total number of attribute values included in subgroup $S$, summed across all covariates $X^1, \ldots, X^m$. Note that there is no penalty for a given attribute if all attribute values are included, since this is equivalent to ignoring the attribute when defining subgroup $S$. The penalty term results in more interpretable subgroups by encouraging the scan either to ignore an attribute (i.e., all values of that attribute are included in the subgroup) or to choose a smaller number of attribute values to include in the subgroup.

We now consider how CBS can efficiently maximize $F(S)$ over subgroups $S$ of the protected class, returning the discovered subgroup $S^*$ and the corresponding score $F(S^*)$, where we aim to identify

$$S^* = \arg\max_S F(S). \tag{2}$$

The scan procedure for CBS takes as inputs a dataset $D_1 = (I, \hat{I}, X)$ consisting of the *observed* event variable $I_i$, the estimated expectation of $I_i$ under the null hypothesis $\hat{I}_i$, as calculated in Section 2.2, and the covariates $X_i$, for each individual in the protected class ($A_i = 1$), along with several parameters: the type of scan (Gaussian or Bernoulli), the direction of bias to scan for (over- or under-estimation bias), complexity penalty, and number of iterations. It then searches for the highest-scoring subgroup (consisting of a non-empty subset of values $V^j$ for each covariate $X^j$), starting with a random initialization on each iteration, and proceeding by *coordinate ascent*.

The coordinate ascent step identifies the highest-scoring non-empty subset of values $V^j$ for a given covariate $X^j$, conditioned on the current subsets of values $V^{-j}$ for all other attributes. As shown in McFowland III et al. (2023), each individual coordinate ascent step can provably find the optimal subset of attribute values while evaluating only $|X^j|$ of the $2^{|X^j|}$ subsets of values, where $|X^j|$ is the arity of covariate $X^j$. This efficient subroutine follows from the fact that the score functions above satisfy the additive linear-time subset scanning property (Neill, 2012; Speakman et al., 2016). The coordinate ascent step is repeated with different, randomly selected covariates until convergence to a local optimum of the score function, and multiple random restarts enable the scan to approach the global optimum. McFowland III et al. (2023) provide sufficient conditions under which this routine will identify the global optimum in the large-sample limit; empirically, the approach converges to near-optimal subgroups while requiring only low-order polynomial time.

For an in-depth, self-contained description of the scan algorithm, including pseudocode (Algorithm 2 in Appendix A.1.2), an analysis of its computational complexity (Appendix A.1.3), and how it exploits an additive property of the score functions to achieve linear-time efficiency for each scan step (Appendix A.1.1), see Appendix A.1. Derivations for $F(S)$ are provided in Appendix A.2.

---

**Algorithm 1** Semi-Synthetic Data, $D_{fair}$

---

1: **Require:** $(X_1, ..., X_m) = X$
2: Randomly pick protected class attribute $A$
3: $X_{\overline{A}} = X \setminus A$
4: $w_j \sim \mathcal{N}(0, 0.2)$ where $\forall j = 1$ **to** $ncols(X_{\overline{A}})$
5: **for** $i = 1$ **to** $nrows(X_{\overline{A}})$ **do**
6:      $L_i^{true} = (\Sigma_{j=1}^{m-1} w_j X_i^{(j)}) + \epsilon_i^{true}$        $\triangleright$ where $\epsilon_i^{true} \sim \mathcal{N}(0, \sigma_{true})$, $L_i^{true}$ represents the true log-odds of a positive outcome for row $i$.
7:      $Y_i \sim \text{Bernoulli}(\sigma(L_i^{true}))$        $\triangleright$ $\sigma$ is the sigmoid function, $Y_i$ is the true outcome for row $i$.
8:      $L_i^{predict} = L_i^{true} + \epsilon_i^{predict}$        $\triangleright$ where $\epsilon_i^{predict} \sim \mathcal{N}(0, \sigma_{predict})$, $L_i^{predict}$ represents the predicted log-odds for row $i$, $\epsilon_i^{predict}$ represents non-systematic errors (random noise) in the predictive model.
9:      $P_i = \sigma(L_i^{predict})$        $\triangleright$ $\sigma$ is the sigmoid function, $P_i$ is the predicted probability of a positive outcome for row $i$.
10:      $P_{i,bin} = \mathbf{1}\{P_i \geq 0.5\}$        $\triangleright$ $P_{i,bin}$ is the predicted recommendation for row $i$ based on thresholding $P_i$.
11: **end for**
12: **return** $D_{fair} = (X_{\overline{A}}, A, L^{true}, L^{predict}, Y, P, P_{bin})$        $\triangleright$ return covariates, protected class attribute, true log-odds, predicted log-odds, outcomes, probabilities, and recommendations

---

## 2.4 Permutation Testing to Evaluate the Statistical Significance of $S^*$

As described in Section 2.3, the scan step returns the detected subgroup $S^*$ that maximizes the score function for a given scan type and parameterization, $S^* = \arg\max_S F(S)$ as shown in Equation 2. The statistical significance ($p$-value) of the discovered subgroup $S^*$ can be obtained by *permutation testing*, which correctly adjusts for the multiple testing resulting from searching over subgroups.

To do so, we generate a large number of simulated datasets under the null hypothesis $H_0$. For each null dataset, we generate new estimates for $\hat{I}$, as described in Section 2.2, and perform the same CBS scan, as described in Section 2.3 (maximizing the log-likelihood ratio score over subgroups, exactly as performed for the original dataset). We then compare the maximum score $F(S^*)$ found for the true dataset to the distribution of maximum scores $F(S^*)$ found for the simulated datasets. To generate each simulated dataset under the null hypothesis, we copy the original dataset and randomly permute the values of $A_i$ (whether or not each individual is a member of the protected class), thus testing the null hypothesis that $A$ is conditionally independent of the event variable $I$. The detected subgroup is significant at level $\alpha$ if its score exceeds the $1 - \alpha$ quantile of the $F(S^*)$ values for the simulated datasets. For a given dataset, the score threshold for significance at a fixed level $\alpha = .05$ will differ for different choices of the sensitive attribute and protected class. Thus, if CBS is used to audit a classifier for possible biases against multiple protected classes, a separate permutation test must be performed for each protected class value.

This permutation testing approach is computationally expensive, multiplying the runtime by the total number of datasets (original and simulated) on which the CBS scan is performed, but it has the benefit of bounding the overall false positive rate (family-wise type I error rate) of the scan while maintaining high detection power. In comparison, the simpler approach of Bonferroni correction also bounds the overall false positive rate, and requires much less runtime, but suffers from dramatically reduced detection power.

Finally, we note that this procedure does not account for additional multiple testing issues which could result if we run several conditional bias scans (e.g., with different choices of the protected class or different group fairness definitions) but wish to bound the total type I error rate across all scans. An additional Bonferroni correction (dividing the p-value threshold for statistical significance by the number of CBS runs, *not* the number of subgroups) can be applied in this case.

## 3 Evaluation

Given the lack of gold standard approaches for evaluating subgroup bias auditing methods, we evaluate the CBS framework through semi-synthetic simulations with the following steps:

Step (A): Given a real-world dataset with attributes $X$, randomly select a protected class $A$ and *generate a semi-synthetic dataset*, referred to as $D_{fair}$, where the predictions, recommendations, and outcomes are conditionally independent of $A$ given $X \setminus A$, i.e., there are *no* sufficiency or separation violations (defined in Section 2.1) pertaining to protected class $A$.

Step (B): Take the unmodified semi-synthetic data, $D_{fair}$, and *inject signal* consistent with a separation or sufficiency violation or base rate shift into a subgroup of protected class $A$, referred to as $S_{bias}$, to generate dataset $D_{inj}$.

Step (C): *Run CBS and benchmark methods* to detect violations pertaining to protected class $A$ for $D_{inj}$ and *measure the accuracy of the detected subgroups*, referred to as $S^*$, compared to the known (injected) biased subgroup, $S_{bias}$.

We will discuss each step, Step (A)-Step (C), in detail below, including the motivating questions that underpin our design choices.

*Step (A) Generate a semi-synthetic dataset:* Using COMPAS data[2] described in Section 4, we use Algorithm 1 to generate a semi-synthetic dataset, $D_{fair}$. Specifically, we randomly select an attribute and value to define the protected class $A$ and remove that attribute from $X$, as shown in Lines 2-3. In Line 4, we draw a weight from a Gaussian distribution, $w_j \sim \mathcal{N}(0, 0.2)$, for each attribute-value of the covariates, excluding $A$. We use these weights, $w_j$, in Line 6, to produce the true log-odds, $L_i^{true}$, of a positive outcome ($Y_i = 1$) for each row $i$ by a linear combination of the attribute values with these weights. Note, in Line 6 of Algorithm 1, we add $\epsilon_i^{true} \sim \mathcal{N}(0, \sigma_{true})$ to each row's true log-odds, $L_i^{true}$, representing variation between rows that arises from external factors (not included in the scan attributes), and is incorporated into the predictive model.[3] Given the true log-odds $L_i^{true}$ of $Y_i = 1$ for each row $i$, in Line 7 we draw each outcome $Y_i$ from a Bernoulli distribution with the corresponding probability, $\text{expit}(L_i^{true})$, which we refer to as the true probabilities. Next, in Lines 8-9, we set each row's predicted probability $P_i = \text{expit}(L_i^{true} + \epsilon_i)$, where $\epsilon_i \sim \mathcal{N}(0, \sigma_{predict})$ represents non-systematic errors (random noise) in the predictive model. Finally, we threshold the probabilities to produce recommendations $P_{i,bin} = \mathbf{1}\{P_i \geq 0.5\}$ for each row $i$ in Line 10.

Algorithm 1 returns the randomly selected protected class attribute, $A$, the original covariates, excluding the protected class membership, $X_{\overline{A}}$, the synthetic true labels, $Y$, the predicted probabilities, $P$, and thresholded probabilities to create recommendations, $P_{i,bin}$. **Importantly**, since $A$ is conditionally independent of the outcomes $Y$, predictions $P$ and recommendations $P_{bin}$ given the observed covariates $X$, by design, this dataset, $D_{fair}$, contains *no* signals indicating separation or sufficiency violations for a subgroup of protected class $A$.

We use default values of $\sigma_{true} = 0.6$ and $\sigma_{predict} = 0.2$ for Algorithm 1, and examine sensitivity to these parameters in Appendix B.4; see Appendix B.2 for a discussion of the impact of $\sigma_{true}$ on sufficiency-based fairness definitions.

*Step (B) Inject signal:* We randomly select a subgroup, $S_{bias}$, of the protected class $A$ of $D_{fair}$ into which we will inject biases or base rate shifts to create $D_{inj}$. We pick $S_{bias}$ by randomly choosing two attributes ($n_{bias} = 2$) and then independently including or excluding each value of those attributes with probability $p_{bias} = 0.5$. (This process is repeated until the resulting subgroup is non-empty.)

We designed the evaluation to address three key questions about the performance of the four CBS variants and benchmark methods:

---

[2]We use the covariates from COMPAS to maintain realistic covariate correlations, but do not use the predictions or outcomes.
[3]Rudin et al. (2020) note that COMPAS relies on up to 137 variables collected from a questionnaire, and we expect that some of these additional variables are correlated with outcomes.

(Q1):     How well do they detect *biases* represented as systematic differences, in the form of injected signals $\mu_{sep}$ and $\mu_{suf}$, between the predicted and true probabilities for the event variable $I$ in subgroup $S_{bias}$ of the protected class $A$?

(Q2):     How do they respond to an *injected base rate shift*, i.e., an equal shift by $\delta$ in the predicted and true probabilities for the event variable $I$ for subgroup $S_{bias}$ of the protected class $A$?

(Q3):     How do the answers to the first two questions, (Q1) and (Q2), vary based on the size of the biased subgroup $S_{bias}$, which is controlled by the values chosen for $n_{bias}$ and $p_{bias}$?

To address (Q1), we inject a bias signal into subgroup $S_{bias}$ of the protected class $A$, keeping the corresponding subgroup of the non-protected class unchanged, in one of two ways, corresponding to separation and sufficiency violations respectively:

bias $\mu_{sep}$:     To produce $D_{inj}$ from $D_{fair}$, for each row of $S_{bias}$, we increase the predicted probability $P_i$ by $\mu_{sep}$ and then recompute the model's recommendation $P_{i,bin}$ by thresholding $P_i$ at 0.5. When $\mu_{sep} > 0$, this creates a signal which is consistent with **separation violations** in the *positive direction* for subgroup $S_{bias}$ of protected class $A$ in $D_{inj}$. Detection results of CBS and benchmark methods for the injected signal of $\mu_{sep}$ to $S_{bias}$ are in the leftmost plot of Figure 1.

bias $\mu_{suf}$:     To produce $D_{inj}$ from $D_{fair}$, for each row of $S_{bias}$, we reduce the true probability by $\mu_{suf}$ and then redraw the outcome $Y_i$. When $\mu_{suf} > 0$, this creates a signal which is consistent with **sufficiency violations** in the *negative direction* for subgroup $S_{bias}$ of protected class $A$ in $D_{inj}$. Detection results of CBS and benchmark methods for the injected signal of $\mu_{suf}$ to $S_{bias}$ are in the rightmost plot of Figure 1.

Both of these injected signals result in a bias where $P$ and $P_{bin}$ overestimate the outcomes ($Y$) for the given subgroup $S_{bias}$ of the protected class $A$ in $D_{inj}$.

To address (Q2), we inject a base rate shift into subgroup $S_{bias}$ of the protected class $A$, keeping the corresponding subgroup of the non-protected class unchanged:

shift $\delta$:   To produce $D_{inj}$ from $D_{fair}$, for each row of $S_{bias}$, we increase *both* the true probabilities and the predicted probabilities of $S_{bias}$ by $\delta$, then redraw outcomes $Y_i$ and recompute recommendations $P_{i,bin}$. For positive $\delta$, this creates a **higher base rate** of a positive outcome for subgroup $S_{bias}$ of the protected class $A$ in $D_{inj}$, as compared to the corresponding subgroup of the non-protected class, while maintaining well-calibrated predictions. Detection results of CBS and benchmark methods for the injected base rate shift of $\delta$ to $S_{bias}$ are in Figure 2.

Importantly, the signals for $\mu_{sep}$, $\mu_{suf}$, and $\delta$ are created by a uniform shift in the true and predicted probabilities, which corresponds to a *non-uniform* shift in the true and predicted log-odds. **This is distinct from the modeling assumption made by CBS**, which assumes (under the alternative hypothesis that bias is present) a constant additive shift in the true or predicted log-odds. By injecting signal in this way, we ensure that our method is robust to non-additive shifts in log-odds. For simulation results that inject bias represented as additive shifts in log-odds, please see Appendix B.4. We observe high consistency between those additional results and the ones presented here.

To address (Q3), we run three experiments ($\mu_{sep} = 0.50$, $\mu_{suf} = 0.50$, and $\delta = 0.25$) while varying the size of $S_{bias}$ in one of two ways:

$n_{bias}$:   We vary the number of attributes, $n_{bias}$, that the attribute-values can be chosen from, between 1 and 4, when randomly selecting $S_{bias}$. Detection results of CBS and benchmark methods when varying $n_{bias}$ with different types of injected signal are in the top row of Figure 3.

$p_{bias}$:   We vary the probability, $p_{bias}$, that each value of the chosen attributes is included in $S_{bias}$. Detection results of CBS and benchmark methods when varying $p_{bias}$ with different types of injected signal are in the bottom row of Figure 3.

Together, the different shifts of $\mu_{sep}$, $\mu_{suf}$, and $\delta$ along with variations in the parameters $n_{bias}$ and $p_{bias}$ used to select $S_{bias}$ result in 1,344 distinct processes that each generate a distinct dataset $D_{inj}$ from the base semi-synthetic dataset, $D_{fair}$, generated in Step (A).

*Step (C) Run CBS and benchmark methods and measure the accuracy of the detected subgroups:* We compare the four variants of CBS to the benchmark methods, GerryFair (Kearns et al., 2018) and MultiAccuracy Boost (Kim et al., 2019a), described in Section 5. For more information about the benchmark methods and the modifications we made to make them more comparable to CBS for these simulations, see Appendix B.1.

After injecting bias into or shifting the base rates of $S_{bias}$ for the protected class $A$ of $D_{fair}$ to create $D_{inj}$, as described in Step (B), we run four variants of CBS (separation scan for recommendations in the positive direction; separation scan for predictions in the positive direction; sufficiency scan for recommendations in the negative direction; and sufficiency scan for predictions in the negative direction) and GerryFair and MultiAccuracy Boost for $D_{inj}$ for the randomly selected protected class attribute, $A$. (Note: We use the same settings for CBS as described in Section 4, with the exception of running all scans with all conditional variable values rather than as value-conditional scans.) Then, we measure the accuracy of a detected subset for each CBS scan and each benchmark method, $S^*$, compared to the ground truth subgroup, $S_{bias}$, using the following metric:

$$\text{accuracy}(S^*) = \frac{|\ S_{bias}\ \cap\ S^*\ |}{|\ S_{bias}\ \cup\ S^*\ |} \tag{3}$$

Equation 3 is the Jaccard similarity between the injected and detected subsets, $S_{bias}$ and $S^*$, respectively. This accuracy measure penalizes both falsely detected unbiased instances and undetected instances affected by bias, making it appropriate for applications where both types of error should be minimized.

Finally, to generate confidence intervals for these simulations, we repeat the full set of experiments 100 times and report the average accuracy of each method across runs, specifically:

- For Step (A), we generate 100 distinct semi-synthetic datasets, $D_{fair}$.

- For each of these 100 datasets of $D_{fair}$, we perform the same 1,344 experiments described in Step (B), each defined by a unique configuration of injected bias. This produces 1,344 distinct biased datasets, $D_{inj}$, per $D_{fair}$.

- For each of the resulting $D_{inj}$, we evaluate the accuracy of the detected subgroup $S^*$ discovered by the four CBS scans and two benchmark methods, as described in Step (C).

This results in performance accuracy scores for all 1,344 experiments, repeated across 100 independently generated $D_{fair}$ datasets, which are then averaged by grouping distinct experiment configurations across all 100 versions of $D_{fair}$ to obtain confidence intervals.

## 3.1 Simulation Results

In Figure 1, which addresses (Q1), we observe that all four variants of CBS are able to detect the injected signal, $\mu_{sep}$ or $\mu_{suf}$, in subgroup $S_{bias}$ of the protected class $A$, with higher detection accuracy (defined in Equation 3) than GerryFair or MultiAccuracy Boost. Sufficiency scans had highest detection accuracy for shifts in true probabilities ($\mu_{suf}$), as shown in the rightmost plot of Figure 1, and separation scans had highest detection accuracy for shifts in predicted probabilities ($\mu_{sep}$), as shown in the leftmost plot of Figure 1. Scans for predictions generally outperformed scans for recommendations, due to the loss of information from binarization of the probabilistic predictions.

Interestingly, sufficiency scan for predictions (but not for recommendations) converged to perfect detection accuracy for $\mu_{sep}$, while separation scans did not converge to perfect detection accuracy for $\mu_{suf}$. Sufficiency scan for predictions is conditioned on a real-valued variable ($P_i$) rather than a binary variable ($P_{i,bin}$ or $Y_i$), allowing for more flexible modeling of $\mathbb{E}[Y \mid P, X]$ and thus greater sensitivity to shifts in predicted probabilities.

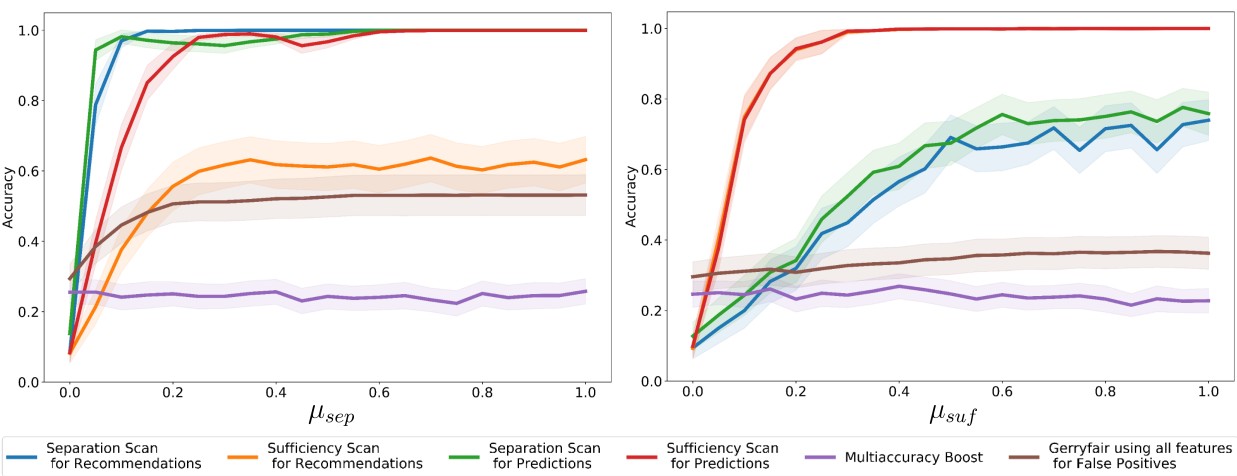

Figure 1: Average accuracy (with 95% CI) as a function of the amount of bias injected into subgroup $S_{bias}$ of the protected class $A$, for four variants of CBS, GerryFair, and MultiAccuracy Boost. Left: increasing predicted probabilities by signal $\mu_{sep}$. Right: decreasing true probabilities by signal $\mu_{suf}$.

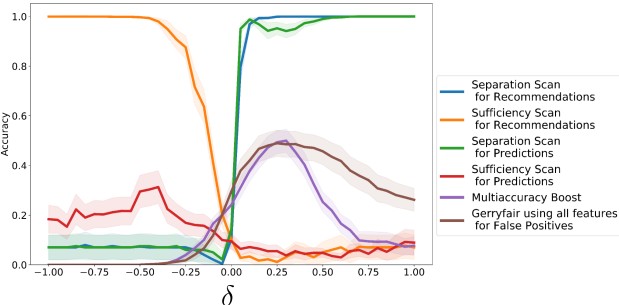

Figure 2: Average accuracy (with 95% CI) as a function of the base rate difference $\delta$ between protected and non-protected class for subgroup $S_{bias}$, for four variants of CBS, GerryFair, and MultiAccuracy Boost. Note that predictions are well calibrated, $\mu_{sep} = \mu_{suf} = 0$, for this set of results.

In Figure 2, which addresses (Q2), shifting the base rate by $\delta$ for subgroup $S_{bias}$ of the protected class $A$ results in separation scans detecting a base rate shift when $\delta > 0$, while sufficiency scans and competing methods are not sensitive to this shift. **This finding aligns with previous research proving that differences in base rates between two populations will result in a higher false positive rate for the population with a higher base rate when using a well-calibrated classifier (Chouldechova, 2017).**

Interestingly, as shown in Figure 2, sufficiency scan for recommendations detects a base rate shift for $\delta \ll 0$. In this case, $\mathbb{E}[Y \mid P_{bin}, X]$ is lower for instances of $S_{bias}$ in the protected class $A$ than for instances with negative recommendations ($P_{bin} = 0$) in the non-protected class. Thus conditioning on the binary indicator $P_{i,bin}$ for this simulation is not sufficient to capture this decrease in the true probabilities, while conditioning on the real-valued prediction $P_i$ allows sufficiency scan for predictions to extrapolate reasonably well to these cases.

In Figure 3, which addresses (Q3), we observe that, when varying $n_{bias}$, which can be observed in the plots contained in the top row of Figure 3, CBS has similar detection accuracy results to the simulations shown in Figures 1 and 2, with separation scans and sufficiency scan for predictions having higher bias detection accuracy when $\mu_{sep} = 0.50$, and sufficiency scans having higher bias detection accuracy when $\mu_{suf} = 0.50$, as

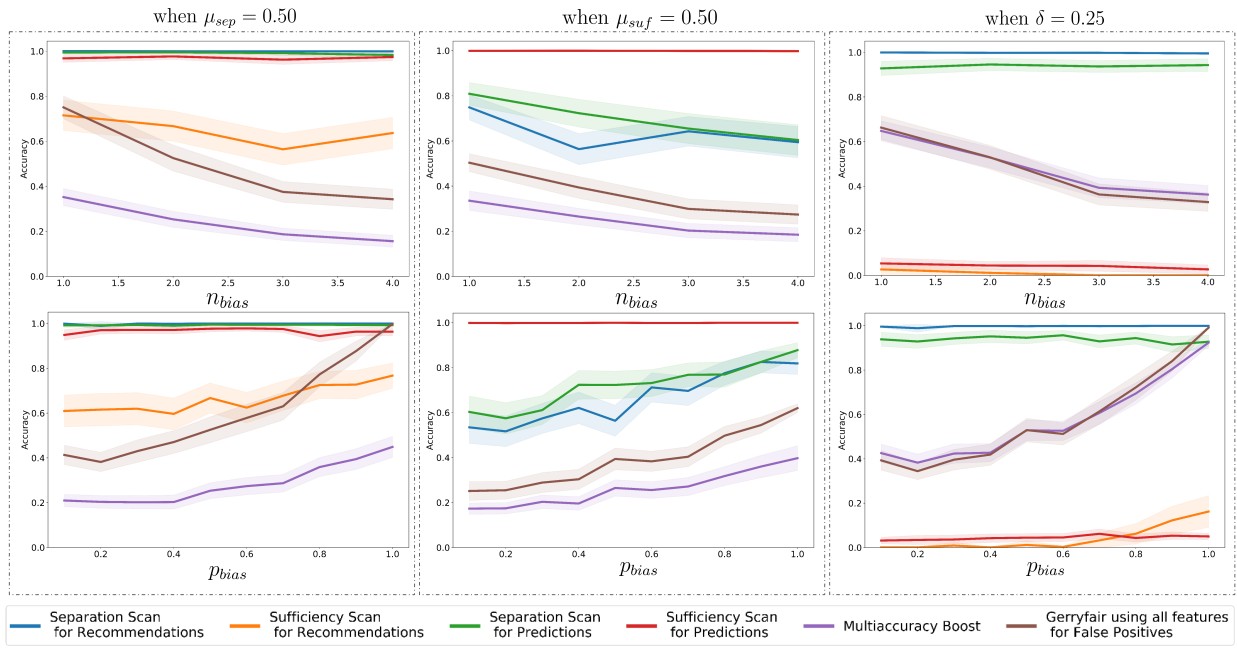

Figure 3: Average accuracy (with 95% CI) for biases and base rate shifts injected into subgroup $S_{bias}$ of the protected class, for CBS, GerryFair, and MultiAccuracy Boost, as a function of varying parameters $n_{bias}$ (top row) and $p_{bias}$ (bottom row). Left: increasing predicted probabilities by $\mu_{sep} = 0.50$. Center: decreasing true probabilities by $\mu_{suf} = 0.50$. Right: base rate difference $\delta = 0.25$, for $\mu_{sep} = \mu_{suf} = 0$.

compared to competing methods across all settings of $n_{bias}$. Interestingly, as observed in the lower leftmost plot of Figure 3, when $\mu_{sep} = 0.50$ and $p_{bias}$ approaches 1 (i.e., more individuals in the protected class $A$ are included in $S_{bias}$), GerryFair has improved bias detection accuracy, approaching that of CBS, but it performs poorly for values of $p_{bias}$ closer to 0. This suggests that CBS is better at detecting smaller, more subtle subgroups $S_{bias}$ than the competing methods.

All fixed hyper-parameter choices for these simulations are moderate values which align with non-edge cases. Additional robustness checks for varying hyper-parameter choices for these simulations are described in Appendix B.4. For estimates of compute power needed for the simulations see Appendix B.5.

## 4 Case Study of COMPAS

The COMPAS algorithm is used in various jurisdictions across the United States as a decision support tool to predict individuals' risk of recidivism. It is commonly used by judges when deciding whether an arrested individual should be released prior to their trial (Angwin et al., 2016b). Following the initial investigation by ProPublica about fairness issues in the risk scores generated by the COMPAS algorithm (Angwin et al., 2016b), ProPublica's COMPAS dataset has been used as a benchmark in the fairness literature. We follow many of the processing decisions made in the initial ProPublica analysis, including removing traffic offenses and defining recidivism as a new arrest within two years of the initial arrest for a defendant (Larson et al., 2016; Larson & Roswell, 2017). After preprocessing the initial data set, we have 6,172 defendants, their gender, race, age (Under 25 or 25+), charge degree (Misdemeanor or Felony), prior offenses (None, 1 to 5, or Over 5), predicted recidivism risk score $s_i$, where $s_i \in \{1, 2, \ldots, 10\}$, and whether they were re-arrested within two years of the initial arrest, where $Y_i = 1$ if individual $i$ was re-arrested and $Y_i = 0$ otherwise. Given that COMPAS only provides risk scores and not predicted probabilities of reoffending, we define each defendant $i$'s predicted probability of reoffending using maximum likelihood estimation:

$$P_i = \frac{\sum_{j=1}^{n} \mathbf{1}\{Y_i = 1 \wedge s_j = s_i\}}{\sum_{j=1}^{n} \mathbf{1}\{s_j = s_i\}} \tag{4}$$

Defendants with COMPAS risk scores of 5+ ($s_i \geq 5$) are considered "high risk" since the COMPAS documentation stipulates careful consideration by supervision agencies for these defendants (Larson et al., 2016). Therefore we define the COMPAS recommendations as $P_{i,bin} = \mathbf{1}\{s_i \geq 5\}$.

We chose the parameters for each of the four variants of CBS scans (values of the event and conditioning variables, $I$ and $C$, respectively, and direction of bias) in order to search for systematic biases in COMPAS predictions and recommendations which *disadvantage* defendants for a given protected class $A$. For the separation scans, we detect positive deviations for the protected class attribute $A$ in the $\mathbb{E}(P \mid Y = 0, X)$, listed as FPE in Table 3, and $\Pr(P_{bin} = 1 \mid Y = 0, X)$, listed as FPR in Table 3, i.e., increase in predicted risk and increase in FPR for non-reoffending defendants, respectively. For the sufficiency scans, we detect a negative deviation for the protected class $A$ in the $\Pr(Y = 1 \mid P, X)$, listed as CAL in Table 3, and $\Pr(Y = 1 \mid P_{bin} = 1, X)$, listed as PPV in Table 3, i.e., decreased probability of reoffending conditional on predicted risk and on being flagged as high-risk, respectively. These choices were made to ensure our ability to verify our findings based on previous research on COMPAS, which commonly focus on similar fairness violations to those used in our case study. With that said, we strongly encourage auditing for predictive biases that affect reoffending defendants and low-risk defendants as well, if using CBS to audit an algorithmic risk assessment tool in practice. For example, auditing for the increased probability of being flagged as high-risk for reoffending defendants could help to uncover subpopulations that are over-prosecuted in comparison to other populations of reoffending defendants. Therefore, expanding the fairness definitions used to audit pre-trial risk assessment tools for biases could have beneficial findings.

For all scans, we use all attributes except for the sensitive attribute when calculating the probability of being a member of the protected class $A$ (for the propensity score weighting in Step 1 and Step 2 of Section 2.2) and when generating the predicted values $\hat{I}$ under the null hypothesis of no bias, $H_0$, in Step 3 of Section 2.2. All scans were run for 500 iterations with a penalty equal to 1.

### 4.1 COMPAS Results

Table 3 contains the detected subgroups $S^*$, and their associated log-likelihood ratio scores $F(S^*)$, and corresponding indicators of statistical significance, found by each of the four variants of CBS, for various choices of the sensitive class attribute: Black, white, female, male, younger (under the age of 25) defendants, older (age 25+) defendants, and defendants with no priors. Please see Section 2.4 for the permutation test procedure used to determine statistical significance of CBS's detected subgroups. For the full set of results for all CBS scans when treating each attribute value as the sensitive class attribute, please see Table 5 in Appendix C.1.2. For a discussion of the benchmark methodologies' results for COMPAS, please reference Appendix C.1.3.

Below are summaries of some of the statistically significant results that CBS found in COMPAS predictions and recommendations displayed in Table 3:

**Racial bias in COMPAS.** Table 3 shows that the separation scans identify statistically significant biases negatively impacting a subgroup of Black defendants, while the sufficiency scans do not. These results support and complement the previous findings by ProPublica (Angwin et al., 2016b) and follow-up analyses (Chouldechova, 2017), which concluded that COMPAS has large error rate disparities which negatively impact Black defendants (corresponding to large log-likelihood ratio scores, $F(S^*)$, for separation scans), and that its predictions are well-calibrated for Black defendants (corresponding to small and statistically insignificant log-likelihood ratio scores for sufficiency scans).

However, CBS's detected subgroup for the two separation scans adds a useful finding to this discussion: **the large FPR disparity of COMPAS against Black defendants is even more significant in the intersectional subgroup of Black males found by CBS's separation scans**. Non-reoffending Black male defendants have an FPR of 0.44, compared to non-reoffending non-Black male defendants' FPR of

| Scan Type | Detected Subgroup ($S^*$) | Log-Likelihood Ratio ($F(S^*)$) | Metric | Observed Metric for Sensitive Detected Subgroup (Num. of Defendants) | Observed Metric for Complement Detected Subgroup (Num. of Defendants) |
|---|---|---|---|---|---|
| \multicolumn{6}{c}{Sensitive Attribute: Black Defendants compared to Non-Black Defendants} ||||||
| Sep. Pred. | Males | **42.4** | FPE | 0.45 (1168) | 0.35 (1433) |
| Sep. Rec. | Males | **102.3** | FPR | 0.44 (1168) | 0.19 (1433) |
| Suff. Pred. | Females | 2.21 | CAL | 0.37 (549) | 0.34 (626) |
| Suff. Rec. | Age 25+ with 0-5 priors | 0.37 | PPV | 0.50 (581) | 0.52 (404) |
| \multicolumn{6}{c}{Sensitive Attribute: White Defendants compared to Non-White Defendants} ||||||
| Sep. Pred. | – | 0.0 | FPE | – | – |
| Sep. Rec. | Females under age 25 with no priors | 2.01 | FPR | 0.71 (31) | 0.56 (70) |
| Suff. Pred. | Under age 25 | 2.36 | CAL | 0.49 (347) | 0.58 (1000) |
| Suff. Rec. | Females under age 25 | 0.41 | PPV | 0.39 (57) | 0.47 (110) |
| \multicolumn{6}{c}{Sensitive Attribute: Female Defendants compared to Male Defendants} ||||||
| Sep. Pred. | White | **1.51** | FPE | 0.38 (312) | 0.35 (969) |
| Sep. Rec. | White | **12.5** | FPR | 0.29 (312) | 0.20 (969) |
| Suff. Pred. | Under age 25 | **18.7** | CAL | 0.38 (246) | 0.60 (1101) |
| Suff. Rec. | Under age 25 | **13.2** | PPV | 0.44 (167) | 0.68 (699) |
| \multicolumn{6}{c}{Sensitive Attribute: Male Defendants compared to Female Defendants} ||||||
| Sep. Pred. | Asian | 0.63 | FPE | 0.30 (22) | 0.22 (1) |
| Sep. Rec. | Asian and Hispanic | **22.5** | FPR | 0.21 (286) | 0.05 (57) |
| Suff. Pred. | Native Americans age 25+ | 31.4 | CAL | 0.14 (7) | 1.00 (2) |
| Suff. Rec. | Native Americans age 25+ | 14.1 | PPV | 0.25 (4) | 1.00 (2) |
| \multicolumn{6}{c}{Sensitive Attribute: Defendants under age 25 compared to Defendants age 25+} ||||||
| Sep. Pred. | All defendants under age 25 | **128.2** | FPE | 0.51 (593) | 0.37 (2770) |
| Sep. Rec. | All defendants under age 25 | **159.3** | FPR | 0.53 (403) | 0.25 (1583) |
| Suff. Pred. | – | 0.0 | CAL | – | – |
| Suff. Rec. | – | 0.0 | PPV | – | – |
| \multicolumn{6}{c}{Sensitive Attribute: Defendants age 25+ compared to Defendants under age 25} ||||||
| Sep. Pred. | – | 0.0 | FPE | – | – |
| Sep. Rec. | Asians arrested on felony charges | 0.74 | FPR | 0.20 (10) | 0.00 (1) |
| Suff. Pred. | Males with 0-5 priors | **92.7** | CAL | 0.35 (2867) | 0.59 (1041) |
| Suff. Rec. | Males with 0-5 priors | **53.0** | PPV | 0.52 (772) | 0.67 (641) |
| \multicolumn{6}{c}{Sensitive Attribute: Defendants with no priors compared to Defendants with 1+ priors} ||||||
| Sep. Pred. | – | 0.0 | FPE | – | – |
| Sep. Rec. | – | 0.0 | FPR | – | – |
| Suff. Pred. | All defendants with no priors | **111.6** | CAL | 0.29 (2085) | 0.54 (4087) |
| Suff. Rec. | All defendants with no priors | **51.0** | PPV | 0.46 (553) | 0.67 (2198) |

Table 3: Select results from CBS scans run on COMPAS data. Sep. Pred. is short for separation scan for predictions in the positive direction where the metric FPE stands for $\mathbb{E}[P \mid Y = 0, X]$. Sep. Rec. is short for separation scan for recommendations in the positive direction where the metric FPR, i.e. false positive rate, is $\Pr(P_{bin} = 1 \mid Y = 0, X)$. Suff. Pred. is short for sufficiency scan for predictions in the negative direction where the metric CAL, i.e. calibration, is $\Pr(Y = 1 \mid P, X)$. Suff. Rec. is short for sufficiency scan for recommendations in the negative direction where the metric PPV, i.e. positive predictive value, is $\Pr(Y = 1 \mid P_{bin} = 1, X)$. The third column contains the log-likelihood ratio, $F(S^*)$ defined in Equation 1, for the detected subgroup, $S^*$, listed in the second column. Note, bold scores of $F(S^*)$ are statistically significant with p-value $<.05$ measured by permutation testing, as described in Section 2.4. For example, for the separation scan for recommendations with Black defendants as the sensitive attribute (second row), Black males had a false positive rate of 0.44 ($n = 1168$) compared to 0.19 for non-Black males ($n = 1433$). Please reference Table 5 in Appendix C.1.2 for the comprehensive set of results for COMPAS.

0.19, whereas non-reoffending Black defendants have an FPR of 0.42, compared to non-reoffending non-Black defendants' FPR of 0.20.

CBS does not find any high-scoring or statistically significant subgroups for white defendants, suggesting that COMPAS predictions do not disadvantage white defendants compared to non-white defendants or subgroups of white defendants compared to their non-white counterparts.

As shown in Table 5 of Appendix C.1.2, sufficiency scans find that Asian defendants arrested on misdemeanor charges have a lower rate of reoffending compared to non-Asian defendants with comparable COMPAS risk scores, and that Hispanic defendants flagged as high-risk by COMPAS have a lower rate of reoffending compared to non-Hispanic defendants flagged as high-risk.

**Gender bias in COMPAS.** While male and female defendants have equal false positive rates overall, separation scan for recommendations detects a statistically significant gender bias: non-reoffending white female defendants have a higher false positive rate than non-reoffending white male defendants (0.29 vs 0.20). Separation scan for predictions detects the same gender bias but to a lesser degree: non-reoffending white females have an expected risk of 0.38, compared to non-reoffending white males with an expected risk of 0.35. Sufficiency scans for both recommendations and predictions detect a statistically significant over-estimation bias for females under the age of 25. 44% of females under the age of 25 who are flagged as "high-risk" by COMPAS reoffend, as compared to a 68% recidivism rate for males under the age of 25 who are flagged as "high-risk" by COMPAS. For both separation and sufficiency scans, thresholding the risk scores to create recommendations results in larger deviations between the subgroups of females and males found by the scans, thereby exacerbating the underlying biases present in the COMPAS risk scores that adversely impact white female defendants and younger female defendants, respectively.

Lastly, separation scan for recommendations finds that non-reoffending Asian and Hispanic male defendants have a statistically significant higher false positive rate of being flagged as high-risk (0.21) in comparison to non-reoffending Asian and Hispanic female defendants (0.05) showing that the COMPAS risk scores have intersectional gender biases that adversely impact different subgroups of male and female defendants.

**Age bias in COMPAS.** Previous research argues that COMPAS relies heavily on the assumption that younger defendants are more likely to reoffend (Rudin et al., 2020) when computing risk scores. Younger defendants have a higher reoffending rate compared to older defendants (0.56 vs. 0.46), and thus, well-calibrated predictions and recommendations would result in younger defendants having higher FPR than older defendants. Our separation scans identify non-reoffending defendants under age 25 as the subgroup with the largest FPR disparity: these defendants have a 53% FPR and average predicted probability of reoffending of 51%, as compared to non-reoffending defendants of age 25+, who have a 25% FPR and average predicted probability of reoffending of 37%. On the other hand, our sufficiency scans identify a large subgroup bias within the protected class of defendants age 25+: older male defendants with 0 to 5 priors have a lower rate of reoffending, as compared to younger male defendants with 0 to 5 priors, both for flagged high-risk defendants (sufficiency scan for recommendations) and for defendants with similar risk scores (sufficiency scan for predictions). This finding highlights the scenario described in Section 1 that CBS is designed to detect: predictions are well-calibrated between older and younger defendants, in aggregate, but not for the detected subgroup of older males with 0 to 5 priors.

**Risk overestimation for defendants with no priors in COMPAS.** Sufficiency scans find that defendants with no priors are disadvantaged by COMPAS because their rate of reoffending is statistically significantly lower than defendants with priors who are assigned similar risk scores by COMPAS. Specifically, the rate of reoffending for defendants with no priors who are flagged as high-risk by COMPAS is 0.46 compared to a 0.67 rate of offending for defendants with 1+ priors who are flagged as high-risk by COMPAS. Relatedly, a sufficiency scan for predictions shows that defendants with no priors assigned similar risk scores to defendants with 1+ priors have a lower rate of reoffending, representing a miscalibration of risk scores for defendants with no priors. While not a demographic *bias* per se, this finding highlights a systematic miscalibration in COMPAS's risk scores that penalizes defendants with no prior convictions.

For our **German Credit Data** case study, see Appendix C.2.

# 5 Related Work

## 5.1 Auditing for Subgroup Biases

Bias Scan (Zhang & Neill, 2016) uses a multidimensional subset scan to search exponentially many subgroups of data, with the goal of identifying the subgroup with the most significantly miscalibrated probabilistic predictions compared to the observed outcomes. Bias Scan *lacks* the functionality of traditional group fairness auditing techniques to define a protected class and to determine whether those individuals are impacted by biased predictions, and is thus limited to asking, "Which subgroup in a dataset has the most miscalibrated predictions?" In contrast, given a protected class $A$, CBS aims to identify biases impacting $A$ or any subgroup of $A$. CBS searches for subgroups within the protected class $A$ with the most significant deviation in their predictions and observed outcomes as compared to the predictions and observed outcomes for the corresponding subgroup of the non-protected class (e.g., a racial bias against Black females as compared to non-Black females). Since Bias Scan solely focuses on the deviation between the predictions and observed outcomes within a subgroup, it would be unable to detect the subgroup with the most significant deviations between the protected and non-protected class unless this subgroup also displays significant miscalibration of predictions. Furthermore, CBS generalizes to separation- and sufficiency-based group fairness metrics, and to probabilistic and binarized predictions. To enable this new functionality, CBS deviates from Bias Scan in substantial ways, including novel preprocessing and estimation techniques (see Section 2.2) and new hypotheses and score functions (see Section 2.3).

GerryFair (Kearns et al., 2018) and MultiAccuracy Boost (Kim et al., 2019a), the two benchmark methods used in Section 3, use an auditor to iteratively detect subgroups while training or correcting a classifier to guarantee subgroup fairness. GerryFair's auditor relies on linear regressions trained to predict differences between the predictions and the observed global error rate of a dataset. MultiAccuracy Boost iteratively forms subgroups by evaluating rows with predictions above and below a threshold to determine which predictions to adjust. CBS's methodology for forming subgroups is more complex because it does not assume a linear relationship between covariates and the difference between the predictions and baseline error rate. For more details about these benchmark methods, reference Appendix B.1. Unlike CBS, these methods provide limited fairness definitions for auditing, and do not return interpretable subgroups that are defined by discrete attribute values of the covariates, but rather identify all rows that have a fairness violation on a given iteration. Since both methods incorporate the predictions in forming subgroups and enable auditing, they are comparable to CBS. In Section 3, we show that CBS has substantially higher bias detection accuracy than GerryFair and MultiAccuracy Boost.

There is other research for subgroup bias auditing which is not directly comparable to CBS. For example, Chouldechova & G'Sell (2017) use a recursive partitioning algorithm to find subgroups where the false positive rate disparity between individuals in the protected and non-protected class differs between two predictive models. In addition to this framework providing limited fairness metrics for auditing, this work is formulated to measure pairwise disparities between two models' predictive performance, whereas CBS separately audits each predictive model's results, making this work ill-suited as a benchmark for CBS.

## 5.2 Learning Fair Classifiers

Several recent quantitative research papers (Bose & Hamilton, 2019; Foulds et al., 2020; Subramanian et al., 2021) have proposed methods for *learning fair classifiers* (as opposed to auditing classifiers) with respect to intersectional and/or contextual biases. In the machine learning literature, Bose & Hamilton (2019) use filtered embeddings to train debiased graph embeddings; Foulds et al. (2020) propose new definitions of intersectional bias and use regularization to train fair classifiers; and Subramanian et al. (2021) propose a classifier trained with bias-constraints and also extend a post-hoc debiasing method called iterative nullspace projection (INLP) to address intersectional bias.

### 5.3 Subgroup Discovery

We present a novel subgroup discovery algorithm to search for predictive bias. Subgroup discovery is a rich research domain. Herrera et al. (2011) provide a comprehensive overview of subgroup discovery, covering various fundamental topics including a sampling of search algorithms and quality measurements. Klösgen (1999) provides a condensed and select overview of the topic of subgroup discovery. Lastly, Leman et al. (2008) present a framework for multi-target attribute subgroup discovery. While this work is significantly different from CBS regarding framing, quality measurements, search algorithms, etc., it provides a useful overview of various considerations of subgroup discovery pertaining to a model's outputs for a given data distribution.

### 5.4 Intersectionality

The concept of intersectionality has a rich history (Crenshaw, 1991a;b; Collins, 2008). Given the importance of intersectional biases, we provide concise resources for the original conceptualizations of "intersectionality". In the sociology literature, intersectionality theory (Crenshaw, 1991a;b; Collins, 2008) describes how individuals' different social positions and identities interact to influence their social experiences, actions, and outcomes. In particular, an individual at the intersection of several minoritized groups may be impacted by multiple historical and continuing systems of power and oppression such as structural racism, sexism, income and wealth disparities, etc.

Throughout this paper we are intentional in distinguishing between intersectional and contextual biases, as defined in Section 1. It is important to note that while an intersectional bias detected by CBS *could* be representative of the distinct type of discrimination described by intersectionality theory in some instances, in others they might not perfectly align. Therefore, an intersectional bias detected by CBS cannot interchangeably be used to refer to instances of sociological intersectionality without further analysis and contextualization.

## 6 Limitations

Our CBS framework is designed to audit a classifier's predictions and recommendations for biases with respect to subgroups of a protected class, whereas competing methods provide mechanisms for *both* auditing and correcting classifiers. Combining auditors with correction and training presents the challenge of how to quantify the inherent trade-offs between performance and fairness when correcting for subgroup biases. Additionally, designing auditors that are linked to correction and training methods reinforces the framing that the primary solution to subgroup biases is to correct the models. Given that fairness is often context-specific, ideas of fairness could differ between stakeholders, and upstream biases exist in data sources used in many socio-technical settings, designing an optimally fair model is not always feasible. **We endorse exploring larger policy shifts (not limited to model correction) to address biases that auditing tools like CBS might unearth that are correlated with broader societal issues.**

CBS is designed to detect biases in the form of group fairness violations represented as conditional independence relationships. While CBS is easily generalizable to other objectives that can be represented as group-level conditional independence relationships, it is less generalizable to other fairness definitions such as individual and counterfactual fairness (Dwork et al., 2012; Kusner et al., 2017).

Our technique for estimating the expectations $\hat{I}$ under the null hypothesis of no bias has the limitation (which is commonly cited in the average treatment effects literature) of only being reliable when using well-specified models for estimating the propensity scores of protected class membership and for estimating $\hat{I}$. Given the consistency of our COMPAS results in Section 4 with other researchers' findings about COMPAS, and the calibration analysis we present in Appendix C.1.1, the process of estimating $\hat{I}$ seems to model the COMPAS data well. With that said, we encourage users of CBS to check estimates of $\hat{I}$ and if necessary, employ procedures common in the econometric literature (Imbens, 2004; Schuler & Rose, 2017) or calibration methods within the computer science literature.

There are various limitations to permutation testing, some of which are discussed in Berger (2000). For CBS specifically, if $\hat{I}$ is poorly estimated during permutation testing, this could result in higher type II errors where CBS is more likely to erroneously fail to reject the null hypothesis $H_0$ of no bias.

Our simulations in Section 3 account for bias in the form of shifts in the predicted and true probabilities (separately and jointly) – which produces predictive and aggregation biases – for a prescribed set of covariate attribute values in the protected class. We provide additional simulations with signal and base rate shifts represented as shifts in the true and predicted log-odds in Appendix B.4. In real-world scenarios, the generative process of bias might differ from the assumptions made in our simulations. Future research could determine and (if necessary) improve CBS's robustness to different generative schemas of bias. While this is a limitation of our simulations, the results of CBS for COMPAS, which is a real-world application where the biases present are not a result of our generative process, are in line with other research about biases in COMPAS and the U.S. criminal justice system at large (Chouldechova & G'Sell, 2017; Everett et al., 2011; Rudin et al., 2020). Additionally, we provide a discussion of the benchmark methodologies' results for COMPAS in Appendix C.1.3 to highlight that CBS has various advantages as an auditor in this real-world application (not restricted by the assumptions used in Section 3) compared to the benchmark methodologies' auditor results.

In relation to our case study of COMPAS presented in Section 4, there have been various critiques of the COMPAS data that range from questioning the accuracy of the sensitive attributes (specifically race), noting missing features in the ProPublica dataset that the COMPAS creators claim are important for score calculations, and most importantly, a lack of evaluation of the biases that exist in the outcome variable of whether a defendant is rearrested within two years of arrest (Fabris et al., 2022). Given that certain types of individuals are arrested at a higher rate than others, the outcome variable of re-arrest most likely under- and over-represents certain subpopulations of defendants. Given the various issues pertaining to COMPAS, our case study results in Section 4 may contain findings that do not perfectly align with the underlying distribution of crime or reoffense rates of the true population that COMPAS purports to capture.

***Critically***, while we use COMPAS as a case study in Section 4 because of its familiarity and supporting research, we want to emphasize the importance of alternative framings of the evaluation of automated decision support tools in the criminal justice systems, such as examining the risks that the system poses to defendants rather than the risk of the defendants to public safety (Mitchell et al., 2021; Meyer et al., 2022; Green, 2020).

## 7   Conclusion

In summary, CBS is a flexible framework that works with group-level fairness definitions that can be represented as conditional independence statements (separation and sufficiency) to detect intersectional and contextual biases within subgroups of the protected class while overcoming some of the issues that arise when only considering fairness violations in aggregate for a single protected attribute value. CBS can discover intersectional and contextual biases in COMPAS scores and German Credit Data, and outperforms similar methods that audit classifiers for subgroup fairness.

### Acknowledgments

This work was partially supported by the National Science Foundation's Program on Fairness in Artificial Intelligence in Collaboration with Amazon under Grant No. IIS-2040898.

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

# A  Methods Appendices for Section 2

## A.1  Fast Subset Scanning for Conditional Bias Scan

In this section, we explain the fast subset scanning (FSS) algorithm used by CBS, with the goal of identifying the subgroup, $S^*$, of the protected class, $A$, with the most biased predictions or recommendations (Neill, 2012). Specifically, we will: (1) introduce FSS using a simplified example, for illustrative purposes, to highlight the computational difficulties inherent in subset scanning; (2) formally define the additive property of the score functions for CBS that enables computationally feasible subset scanning in Appendix A.1.1; and (3) provide an implementation of FSS for CBS in the form of pseudocode in Appendix A.1.2.

Let us assume a dataset consisting of *only* individuals in the protected class ($A = 1$), denoted as $Q = \{(X^1, I, \hat{I})\} = \{(X_i^1, I_i, \hat{I}_i)\}_{i=1}^n$, that contains values of the event variable $I_i$, estimates $\hat{I}_i$ of the expected value of the event variable under the null hypothesis of no bias, and a single categorical covariate attribute $X_i^1$ for each individual $i$ in protected class $A$. For concreteness, we perform a sufficiency scan for predictions, therefore, the event variable $I_i$ is the observed binary outcome $Y_i$ for individual $i$, and the corresponding $\hat{I}_i$ is the estimated $\Pr(Y_i = 1 \mid P_i, X_i)$ under the null hypothesis, $H_0$, that $Y \perp A \mid (P, X)$. $S$ refers to a subgroup of $Q$, which in our simple example is a non-empty subset of values for attribute $X^1$. Since our event variable is binary, we use the Bernoulli likelihood function, shown in the bottom half of Table 2, to represent the hypotheses in the score function, $F(S)$, used to determine the level of anomalousness of a subgroup $S$ of $Q$.

In the worst-case scenario, $X^1$ would be a categorical variable where each row of $Q$ holds a distinct value, meaning that there are $n$ unique values for $X^1$. If we were to score all of the possible $S \subseteq Q$ using $F(S)$, this method would have a runtime of $O(2^n)$, which would be computationally infeasible. To overcome this computational barrier, FSS relies on its score functions, $F(S)$, being a part of an efficiently optimizable class of functions with the goal of finding the most anomalous subset $S^* = \arg\max_{S \subseteq Q} F(S)$ without the need to evaluate all of the subsets of $Q$. The property that determines if a function is a part of this class that enables fast subset scanning is called Additive Linear-Time Subset Scanning (ALTSS) (Speakman et al., 2016) and is formally defined in Section A.1.1. Informally, if $F(S)$ can be represented as an additive set function over all instances $i \in S$ when conditioning on the free parameter ($q$ for the Bernoulli distribution or $\mu$ for the Gaussian distribution, where $q$ and $\mu$ are defined in Table 2), it satisfies the ALTSS property (Speakman et al., 2016).

To explore how FSS exploits the ALTSS property for computationally efficient subset scanning, assume that the categorical covariate $X^1$ for each individual $i$ can only be equal to one of four values, $X_i^1 \in \{a, b, c, d\}$. FSS constructs a subset for each distinct attribute value of $X^1$ such that $S_a = \{i \in Q : X_i^1 = a\}$, $S_b = \{i \in Q : X_i^1 = b\}$, $S_c = \{i \in Q : X_i^1 = c\}$, $S_d = \{i \in Q : X_i^1 = d\}$. Since we are using the likelihood function for the Bernoulli distribution for $F(S)$, $F(S)$ is a concave function of the free parameter $q$, and for illustrative purposes, we will assume that $\max_q F(S) > 0$ for all subsets $S_a, S_b, S_c$ and $S_d$. Therefore, for each subset $S_a, S_b, S_c$ and $S_d$, $F(S)$ is a function over the domain of $q$, where as $q$ increases from $-\infty$, $F(S)$ eventually equals 0 and then the global maximum for $F(S)$ for that given subset, and then starts decreasing until it again reaches a point where $F(S) = 0$, and then remains negative as $q$ approaches $\infty$. FSS identifies three $q$ values for each subset, $S \in \{S_a, S_b, S_c, S_d\}$:

1. The first value of $q$ where $F(S) = 0$ as $q$ increases from $-\infty$ to $\infty$, which we will refer to as $q_{min}$.

2. The second value of $q$ where $F(S) = 0$ as $q$ increases from $-\infty$ to $\infty$, which we will refer to as $q_{max}$.

3. The value of $q$ for $\arg\max_q F(S)$, which we will refer to as $q_{\text{MLE}}$.

Each distinct $q_{min}$ and $q_{max}$ value for subsets ($S_a$, $S_b$, $S_c$, $S_d$) is a value of $q$ where the score function $F(S)$ becomes negative or positive for at least one of these four subsets. By sorting all of the distinct $q_{min}$ and $q_{max}$ values across all the subsets ($S_a$, $S_b$, $S_c$, $S_d$) in ascending order, we construct a list of $q$ values, $\{q_{(1)}, ..., q_{(m)}\}$, where each pair of adjacent values, $q_{(k)}$ and $q_{(k+1)}$, represents an interval of the $q$ domain, $(q_{(k)}, q_{(k+1)})$, for which each subset $S \in \{S_a, S_b, S_c, S_d\}$ has either $F(S) > 0$ for the entire interval or $F(S) < 0$ for the entire interval. For each interval, $(q_{(k)}, q_{(k+1)})$, we perform the following:

1. Find the midpoint of the interval (average of $q_{(k)}$ and $q_{(k+1)}$), which we refer to as $q_k^{\text{mid}}$.

2. Create a new subset $S_k^{\text{aggregate}}$ by aggregating all subsets $S \in \{S_a, S_b, S_c, S_d\}$ where the subset's $q_{min} < q_k^{\text{mid}}$ and the subset's $q_{max} > q_k^{\text{mid}}$, i.e., $F(S) > 0$ when $q = q_k^{\text{mid}}$ and therefore for the entire interval $(q_{(k)}, q_{(k+1)})$. Since the score function is additive, conditioned on $q$, we know that a subset $S$ will make a positive contribution to the score $F(S_k^{\text{aggregate}})$ if and only if $F(S) > 0$ for that value of $q$. Thus, we know that the highest scoring subset $S_k^{\text{aggregate}}$ for that interval $[q_{(k)}, q_{(k+1)}]$ contains all and only those subsets $S$ with $F(S) > 0$ at $q = q_k^{\text{mid}}$.

3. Find the maximum likelihood estimate of $q$, $q_{\text{MLE}}^{\text{aggregate}} = \arg\max_q F(S_k^{\text{aggregate}})$, and the corresponding score $F(S_k^{\text{aggregate}})$.

The aggregate subset, $S_k^{\text{aggregate}}$, with the highest score for $F(S)$ using its associated $q_{\text{MLE}}^{\text{aggregate}}$ is the most anomalous subset when considering subsets formed by combinations of different attribute-values of $X^1$.

For our simplified example, there are at most 8 distinct $q_{min}$ or $q_{max}$ values from the four subsets ($S_a$, $S_b$, $S_c$, $S_d$), and thus at most 7 distinct intervals $(q_{(k)}, q_{(k+1)})$ that must be considered. For a given interval, we need to evaluate only a single subset $S_k^{\text{aggregate}}$, and thus, only 7 of the 15 non-empty subsets of $\{S_a, S_b, S_c, S_d\}$. More generally, if $n$ is the arity (number of attribute values) of categorical attribute $X^1$, at most $2n - 1$ of the $2^n - 1$ non-empty subsets of attribute values must be evaluated to identify the highest-scoring subgroup.

The scenario where the covariates consist of a single categorical attribute is a simplified example, where only a single iteration of FSS is needed to find the optimal subset, $S^*$, of $Q$. When there are two or more attributes for the covariates, multiple iterations of FSS must be performed to approach the optimal subset. On each iteration the following is performed:

1. We define an initial subset, $S_{temp}$ where:

   (a) If it is the first iteration, all of the attribute values for each attribute are included in $S_{temp}$.
   (b) Otherwise, a random subset of attribute values for each attribute are chosen to be included in $S_{temp}$.

2. For each attribute $X^i$, in random order, we construct subsets by partitioning $S_{temp}$ by the distinct attribute values of $X^i$, form intervals across the domain of $q$ for $F(S)$, and then assemble and score the subsets for each interval (as described above). $S_{temp}$ is updated as higher scoring subsets using $F(S)$ are found. Therefore, when an attribute is evaluated, $S_{temp}$ contains only rows of $Q$ that fit the found criteria (in the form of attribute values) from previously evaluated attributes, excluding the attribute currently under consideration. This iterative ascent procedure is repeated until convergence.

Multiple iterations are performed with the final detected subset being the subset with the highest score using $F(S)$ found across all iterations, $S^*$. For the pseudocode of FSS for CBS, please see Algorithm 2. The returned results from FSS are: (1) the detected subset, $S^*$, in the form of attribute-values that form the criteria for the subgroup in the protected class with the highest score $F(S)$; (2) the parameter $q$ or $\mu$ that maximizes $F(S^*)$; and (3) and the score $F(S^*)$ given the parameter $q$ or $\mu$.

### A.1.1 Formal Definition of Additive Linear-Time Subset Scanning Property (ALTSS)

Below we provide a formal definition of the Additive Linear-Time Subset Scanning Property. The score functions, $F(S)$, used to evaluate subgroups are a log-likelihood ratio formed from two different hypotheses whose likelihoods are modeled by likelihood functions for either the Bernoulli distribution or Gaussian distribution, both of which satisfy the Additive Linear-time Subset Scanning Property (Speakman et al., 2016; Zhang & Neill, 2016).

**Definition A.1** (Additive Linear-time Subset Scanning Property). A function, $F : S \times \theta \to \mathbb{R}_{\geq 0}$, that produces a score for a subset $S \subseteq D$, where $D$ is a set of data and $\theta = \arg\max_\theta F(S \mid \theta)$, satisfies the Additive Linear-time Subset Scanning Property if $F(S \mid \theta) = \sum_{s_i \in S} F(s_i \mid \theta)$ where $s_i$ is a subset of $S$ and $\forall s_i, s_j \in S$, where $s_i \neq s_j$, we have $s_i \cap s_j = \emptyset$.

We refer to the score functions, $F(S)$, contained in the rightmost column of Table 2 as $F(S \mid \mu)$ for the score functions that use the Gaussian likelihood function to form hypotheses and $F(S \mid q)$ for the score functions that use the Bernoulli likelihood function to form hypotheses. $F(S \mid q)$ contains a summation, $\sum_{i \in S}(I_i \log q - \log(q\hat{I}_i - \hat{I}_i + 1))$, that is the sum of individual-specific values derived from $I_i$, $\hat{I}_i$, and $q$. Given that each individual is distinct, $F(S \mid q) = \sum_{i \in S} F(s_i \mid q)$, where $s_i$ is the subset of $S$ that contains only individual $i$, satisfies the ALTSS property. Similarly, $F(S \mid \mu)$ contains a summation, $\sum_{i \in S} \Delta_i$, that is the sum of individual-specific values $\Delta_i$ derived from $I_i$, $\hat{I}_i$, and $\mu$. Therefore $F(S \mid \mu) = \sum_{s_i \in S} F(s_i \mid \mu)$, where $s_i$ is the subset of $S$ that contains only individual $i$, satisfies the ALTSS property.

### A.1.2 Pseudocode of Fast Subset Scan Algorithm for Conditional Bias Scan

Algorithm 2 is the pseudocode for the Fast Subset Scan (FSS) algorithm used in the CBS framework (Neill, 2012). The FSS algorithm aims to find the subgroup, $S^*$, with the most anomalous signal for the observed event variable, $I$ (i.e., the highest score $F(S^*)$) in a dataset, $D$. For CBS, this signal is in the form of a bias (according to one of the fairness definitions in Table 1) against members of subgroup $S^*$ of the protected class ($A = 1$). The dataset passed to the FSS algorithm by CBS contains only individuals $i$ in the protected class, and FSS compares their values of the event variable $I_i$ to the estimated expectations $\hat{I}_i$ under the null hypothesis of no bias. The method for estimating $\hat{I}_i$ is performed prior to running the FSS scan step and is described in detail in Section 2.2.

At the initialization of FSS, placeholder variables are created that will hold the highest-scoring (most anomalous) subset ($S^*$), and $S^*$'s corresponding information ($\theta^*$, $Score^*$), found across all iterations (Lines 1-3). At the beginning of an iteration, a random subset is picked (set of attribute-values) as the starting subset, $S_{temp}$, with the exception of the first iteration where the starting subset includes all attribute values, as shown in the if-else statement starting on Line 5. For each iteration of this algorithm, we repeatedly choose a random attribute to scan (i.e., we scan over subsets of its attribute values) as shown in Lines 14-15, until convergence (i.e., when all attributes have been scanned without increasing the score $F(S_{temp})$).

For each attribute $X_{temp}$ to be scanned, for each of its attribute values, $X_{temp_i}$, we score the subset $S_{X_{temp_i}}$ containing only the records with the given value of that attribute ($X_{temp} = X_{temp_i}$), and matching subset $S_{temp}$ on all other attributes in $X$. We write this as $S_{X_{temp_i}} \leftarrow S_{temp}^{relaxed} \cap \{i \in D : X_{temp} = X_{temp_i}\}$, in Line 18, where $S_{temp}^{relaxed}$ is the relaxation of subset $S_{temp}$ to include all values for attribute $X_{temp}$. Along with scoring this attribute-value subset $S_{X_{temp_i}}$, we find the two values of $\theta$ where $F(S_{X_{temp_i}}) = 0$, $\theta_{min_i}$ and $\theta_{max_i}$, and the $\theta$ that maximizes $F(S_{X_{temp_i}})$, $\theta_{MLE_i}$, with the exception of attribute-value subsets $S_{X_{temp_i}}$ that are not positive for any value of $\theta$. This is shown in the for-loop in Lines 16-21.

Line 21 states that $\theta_{min_i}$ and $\theta_{max_i}$ must be adjusted according to the direction of the scan to enforce that the found parameters $\theta_{min_i}$ and $\theta_{max_i}$ adhere to the restrictions set by the direction of the scan. The constraints necessary for the scans to detect biases in the positive and negative directions are fully specified in Table 2. For positive scans that have score functions that utilize the Gaussian likelihood function to form hypotheses, $\theta_{min_i} = \max(0, \theta_{min_i})$ and for negative scans that utilize the Gaussian likelihood function, $\theta_{max_i} = \min(0, \theta_{max_i})$. For positive scans that have score functions that utilize the Bernoulli likelihood function to form hypotheses, $\theta_{min_i} = \max(1, \theta_{min_i})$ and for negative scans that utilize the Bernoulli likelihood function, $\theta_{max_i} = \min(1, \theta_{max_i})$. Attribute-value subsets $S_{X_{temp_i}}$ should not be considered when choosing subsets for $S^{\text{aggregate}}$ for positive scans where $\theta_{max_i} < 0$ or $\theta_{max_i} < 1$ for scans using the Gaussian likelihood function or Bernoulli likelihood function in $F(S)$, respectively. Conversely, attribute-value subsets $S_{X_{temp_i}}$ should not be considered when choosing subsets for $S^{\text{aggregate}}$ for negative scans where $\theta_{min_i} > 0$ or $\theta_{min_i} > 1$ for scans using the Gaussian likelihood function or Bernoulli likelihood function in $F(S)$, respectively.

We sort the $\theta_{min_i}$ and $\theta_{max_i}$ values found across all the attribute values of the attribute we are scanning in ascending order in Line 23. These form a list of intervals over the domain of $\theta$. For each interval, we calculate a midpoint of that interval, and aggregate all the attribute-value subsets that have a positive score, $F(S)$, when $\theta$ equals the midpoint of that interval in Lines 30-33. If the aggregated subset of attribute values with the maximum score across all the intervals is greater than the score of $S_{temp}$, we update $S_{temp}$ and all of its accompanying information ($\theta_{temp}$, $Score_{temp}$) to equal the maximum-scoring subset of aggregated attribute-values across all the intervals and its accompanying information, as shown in Lines 43-48. Therefore,

---

**Algorithm 2** Fast Subset Scan (FSS) for Conditional Bias Scan

---

**Require:** $n_{iters} > 0, (X_i, \hat{I}_i, I_i) \, \forall i \in D$ where $A_i = 1, direction \in \{\text{positive}, \text{negative}\}$

1: $S^* \leftarrow \{\}$
2: $Score^* \leftarrow -\infty$
3: $\theta^* \leftarrow -\infty$
4: **for** $j \leftarrow 1 \dots n_{iters}$ **do**
5:      **if** $j == 1$ **then**
6:          $S_{temp} \leftarrow$ all attribute-values for each attribute in $X$
7:      **else**
8:          $S_{temp} \leftarrow$ random nonempty subset of attribute-values for each attribute in $X$
9:      **end if**
10:      $\theta_{temp} \leftarrow \arg\max_\theta (F(S_{temp} \mid \theta))$
11:      $Score_{temp} \leftarrow F(S \mid \theta_{temp})$
12:      $n_{attributes} \leftarrow$ number of attributes in $X$
13:      $n_{scanned} \leftarrow 0$            ▷ mark all attributes as unscanned
14:      **while** $n_{scanned} < n_{attributes}$ **do**
15:          $X_{temp} \leftarrow$ randomly selected attribute that is marked as unscanned
16:          **for** $X_{temp_i} \in X_{temp}$ **do**            ▷ for all attribute-values in $X_{temp}$
17:              $S_{X_{temp_i}} \leftarrow S_{temp}^{relaxed} \cap \{i \in D : X_{temp} = X_{temp_i}\}$      ▷ see Appendix A.1.2 for definition of $S_{temp}^{relaxed}$
18:              $\theta_{min_i}, \theta_{max_i} \leftarrow \arg_\theta (F(S_{X_{temp_i}} \mid \theta) = 0)$      ▷ exception noted in Appendix A.1.2
19:              $\theta_{MLE_i} = \arg\max_\theta (F(S_{X_{temp_i}} \mid \theta))$
20:              $Score_i \leftarrow F(S_{temp_i} \mid \theta_{MLE_i})$
21:              Adjust $\theta_{min_i}$ and $\theta_{max_i}$ depending on the *direction* of scan      ▷ explained in text of Appendix A.1.2
22:          **end for**
23:          $\theta_{intervals} \leftarrow \{\theta_{min_i}, \theta_{max_i} \forall X_{temp_i} \in X_{temp}\}$ in ascending order      ▷ all values of $\theta$ where $F(S) = 0 \, \forall X_{temp_i} \in X_{temp}$, indexed by $\theta_{(k)}$ below
24:          $Score_{interval} \leftarrow -\infty$
25:          $S_{interval} \leftarrow \{\}$
26:          $\theta_{interval} \leftarrow -\infty$            ▷ not to be confused with $\theta_{intervals}$
27:          **for** $k \leftarrow 1 \dots length(\theta_{intervals}) - 1$ **do**
28:              $S_k^{\text{aggregate}} \leftarrow \{\}$
29:              $\theta_k^{\text{mid}} \leftarrow \frac{\theta_{(k)} + \theta_{(k+1)}}{2}$
30:              **for** $X_{temp_i} \in X_{temp}$ **do**
31:                  **if** $Score_i > 0$ and $\theta_{min_i} < \theta_k^{\text{mid}}$ and $\theta_{max_i} > \theta_k^{\text{mid}}$ **then**
32:                      $S_k^{\text{aggregate}} \leftarrow S_k^{\text{aggregate}} \cup S_{X_{temp_i}}$
33:                  **end if**
34:              **end for**
35:              $\theta_k^{\text{aggregate}} \leftarrow \arg\max_\theta (F(S_k^{\text{aggregate}} \mid \theta))$
36:              $Score_k^{\text{aggregate}} \leftarrow F(S_k^{\text{aggregate}} \mid \theta_k^{\text{aggregate}})$
37:              **if** $Score_k^{\text{aggregate}} > Score_{interval}$ **then**
38:                  $Score_{interval} \leftarrow Score_k^{\text{aggregate}}$
39:                  $S_{interval} \leftarrow S_k^{\text{aggregate}}$
40:                  $\theta_{interval} \leftarrow \theta_k^{\text{aggregate}}$
41:              **end if**
42:          **end for**

---

---

43:         **if** $Score_{temp} < Score_{interval}$ **then**

44:            $Score_{temp} \leftarrow Score_{interval}$

45:            $S_{temp} \leftarrow S_{interval}$

46:            $\theta_{temp} \leftarrow \theta_{interval}$

47:            $n_{scanned} \leftarrow 0$                          ▷ mark all attributes as unscanned

48:         **end if**

49:         $n_{scanned} \leftarrow n_{scanned} + 1$                ▷ mark attribute $X_{temp}$ as scanned

50:      **end while**

51:      **if** $Score^* < Score_{temp}$ **then**

52:         $Score^* \leftarrow Score_{temp}$

53:         $S^* \leftarrow S_{temp}$

54:         $\theta^* \leftarrow \theta_{temp}$

55:      **end if**

56:  **end for**

57:  **return** $S^*, Score^*, \theta^*$

---

$S_{temp}$ is continuously updated as higher scoring subsets are found as we scan over all the attributes and their attribute values.

At the end of an iteration, if the found subset, $S_{temp}$, has a higher score than the global maximum scoring subset $S^*$, then $S^*$ and its accompanying information ($\theta^*$, $Score^*$) are replaced with $S_{temp}$ and $S_{temp}$'s accompanying information, as shown in Lines 51-55. Once all the iterations have completed, the subset with the maximum score found across all iterations is returned, $S^*$, with its score , $Score^* = F(S^* \mid \theta^*)$, and accompanying $\theta^*$ parameter in Line 57.

McFowland III et al. (2023) show that a similar multidimensional scan algorithm, used for heterogeneous treatment effect estimation, will converge with high probability to a near-optimal subset when run with multiple iterations.

### A.1.3 Computational Complexity of the Fast Subset Scan Algorithm

The computational complexity of the fast subset scan can be expressed as the product of the number of iterations ($n_{iters}$), the number of covariate attributes ($m$), the number of passes through the attributes required for convergence ($Z$), and the time needed to process a single attribute (Algorithm 2, Lines 16-49). $Z$ is generally small (no more than 10-20). Optimizing over all non-empty subsets of values for a given attribute, conditional on the current subsets of values for all other attributes, requires aggregation of data for the subsets $S_{X_{temp_i}}$ (Line 17), computing and sorting the $\theta_{min_i}$ and $\theta_{max_i}$ values (Lines 18-23), and scoring the aggregated subsets (Lines 27-42). Aggregation requires a pass through the data and is thus an $O(n)$ operation, where $n$ is the number of data elements. The remaining steps require $O(|V| \log |V|)$ time, where $|V|$ is the arity (number of attribute values) for the current attribute $X_{temp}$. Thus the total complexity can be written as $O(n_{iters} m Z(n + \overline{|V| \log |V|}))$, where $\overline{|V| \log |V|}$ is averaged over all attributes.

### A.2 Derivation of Score Functions in Table 2

In this section, we provide derivations of the Gaussian and Bernoulli log-likelihood ratio score functions $F(S)$ shown in Table 2.

### A.2.1 Derivation of Gaussian Score Function

For the Gaussian score function, we compare the null hypothesis $H_0$: $\Delta_i \sim N(0, \sigma)$ to the alternative hypothesis $H_1(S)$: $\Delta_i \sim N(\mu, \sigma)$, $\forall i \in S$, and $\Delta_i \sim N(0, \sigma)$, $\forall i \notin S$. Here, $\mu$ is the maximum likelihood value of the free parameter (additive shift in log-odds) for the alternative hypothesis $H_1(S)$, conditional on the constraints for $\mu$ listed in Table 2. Specifically, for over-estimation bias, $\mu$ is constrained to be less than 0,

while for under-estimation bias, $\mu$ is constrained to be greater than 0. We can then write $F(S)$ as follows, where $D$ represents a data set and $S$ represents a subgroup, $S \subseteq D$, as defined in Section 2:

$$
\begin{aligned}
F(S) &= \log \frac{\Pr(D \mid H_1(S))}{\Pr(D \mid H_0)} \\
&= \log \frac{\max_\mu \Pr(D \mid H_1(S); \mu)}{\Pr(D \mid H_0)} \\
&= \log \frac{\max_\mu \prod_{i \in S} \frac{1}{\sigma\sqrt{2\pi}} \exp\left(-\frac{(\Delta_i - \mu)^2}{2\sigma^2}\right) \prod_{i \notin S} \frac{1}{\sigma\sqrt{2\pi}} \exp\left(-\frac{\Delta_i^2}{2\sigma^2}\right)}{\prod_{i \in D} \frac{1}{\sigma\sqrt{2\pi}} \exp\left(-\frac{\Delta_i^2}{2\sigma^2}\right)} \\
&= \log \frac{\max_\mu \prod_{i \in S} \frac{1}{\sigma\sqrt{2\pi}} \exp\left(-\frac{(\Delta_i - \mu)^2}{2\sigma^2}\right)}{\prod_{i \in S} \frac{1}{\sigma\sqrt{2\pi}} \exp\left(-\frac{\Delta_i^2}{2\sigma^2}\right)} \\
&= \log \max_\mu \prod_{i \in S} \exp\left(\frac{\Delta_i^2 - (\Delta_i - \mu)^2}{2\sigma^2}\right) \\
&= \max_\mu \sum_{i \in S} \left(\frac{\Delta_i^2 - (\Delta_i^2 + \mu^2 - 2\mu\Delta_i)}{2\sigma^2}\right) \\
&= \max_\mu \frac{2\mu\left(\sum_{i \in S} \Delta_i\right) - |S|\mu^2}{2\sigma^2}.
\end{aligned}
$$

The maximum likelihood value of $\mu$ can be obtained by setting the first derivative of $F(S)$ with respect to $\mu$ to zero and enforcing the constraints on $\mu$ listed above:

$$
\mu = \begin{cases} \frac{\sum_{i \in S} \Delta_i}{|S|} & \text{if } \sum_{i \in S} \Delta_i < 0 \text{ and we are searching for over-estimation bias;} \\ \frac{\sum_{i \in S} \Delta_i}{|S|} & \text{if } \sum_{i \in S} \Delta_i > 0 \text{ and we are searching for under-estimation bias;} \\ 0 & \text{otherwise.} \end{cases}
$$

Finally, plugging in this value of $\mu$ into the equation for $F(S)$, we obtain:

$$
F(S) = \begin{cases} \frac{\left(\sum_{i \in S} \Delta_i\right)^2}{2\sigma^2 |S|} & \text{if } \sum_{i \in S} \Delta_i < 0 \text{ and we are searching for over-estimation bias;} \\ \frac{\left(\sum_{i \in S} \Delta_i\right)^2}{2\sigma^2 |S|} & \text{if } \sum_{i \in S} \Delta_i > 0 \text{ and we are searching for under-estimation bias;} \\ 0 & \text{otherwise.} \end{cases}
$$

### A.2.2 Derivation of Bernoulli Score Function

For the Bernoulli score function, we compare the null hypothesis $H_0$: $odds(I_i) = \frac{\hat{I}_i}{1 - \hat{I}_i}$ to the alternative hypothesis $H_1(S)$: $odds(I_i) = q\frac{\hat{I}_i}{1 - \hat{I}_i}$, $\forall i \in S$, and $odds(I_i) = \frac{\hat{I}_i}{1 - \hat{I}_i}$, $\forall i \notin S$. Here, $q$ is the maximum likelihood value of the free parameter (multiplicative shift in odds) for the alternative hypothesis $H_1(S)$, conditional on the constraints for $q$ listed in Table 2. (As discussed in Section 2.2, $\hat{I}$ is the estimate of $\Pr(I_i = 1 \mid X_i, C_i)$ under the condition that $I \perp A \mid (C, X)$. As defined in Section 2.1, the variables represented by $I$ and $C$ change depending on the scan type.) Equivalently, we can write the null hypothesis as $H_0$: $I_i \sim \text{Bernoulli}(\hat{I}_i)$, and the alternative hypothesis as $H_1(S)$: $I_i \sim \text{Bernoulli}\left(\frac{q\hat{I}_i}{q\hat{I}_i - \hat{I}_i + 1}\right)$, $\forall i \in S$, and $I_i \sim \text{Bernoulli}(\hat{I}_i)$, $\forall i \notin S$. For over-estimation bias, $q$ is constrained to be less than 1, and for under-estimation bias, $q$ is constrained to

be greater than 1. We can then write:

$$
\begin{aligned}
F(S) &= \log \frac{\Pr(D \mid H_1(S))}{\Pr(D \mid H_0)} \\
&= \log \frac{\max_q \Pr(D \mid H_1(S); q)}{\Pr(D \mid H_0)} \\
&= \log \frac{\max_q \prod_{i \in S} \left(\frac{q\hat{I}_i}{q\hat{I}_i - \hat{I}_i + 1}\right)^{I_i} \left(\frac{1 - \hat{I}_i}{q\hat{I}_i - \hat{I}_i + 1}\right)^{1 - I_i} \prod_{i \notin S} \left(\hat{I}_i\right)^{I_i} \left(1 - \hat{I}_i\right)^{1 - I_i}}{\prod_{i \in D} \left(\hat{I}_i\right)^{I_i} \left(1 - \hat{I}_i\right)^{1 - I_i}} \\
&= \log \frac{\max_q \prod_{i \in S} \left(\frac{q\hat{I}_i}{q\hat{I}_i - \hat{I}_i + 1}\right)^{I_i} \left(\frac{1 - \hat{I}_i}{q\hat{I}_i - \hat{I}_i + 1}\right)^{1 - I_i}}{\prod_{i \in S} \left(\hat{I}_i\right)^{I_i} \left(1 - \hat{I}_i\right)^{1 - I_i}} \\
&= \log \max_q \prod_{i \in S} \left(\frac{q}{q\hat{I}_i - \hat{I}_i + 1}\right)^{I_i} \left(\frac{1}{q\hat{I}_i - \hat{I}_i + 1}\right)^{1 - I_i} \\
&= \max_q \sum_{i \in S} (I_i \log(q) - \log(q\hat{I}_i - \hat{I}_i + 1)).
\end{aligned}
$$

As in the Gaussian case, the maximum likelihood value of $q$ can be obtained by setting the first derivative of $F(S)$ with respect to $q$ to zero, obtaining the equation:

$$
\sum_{i \in S} I_i = \sum_{i \in S} \frac{q\hat{I}_i}{q\hat{I}_i - \hat{I}_i + 1}.
$$

Unlike the Gaussian case, there is no closed-form solution. However, the right-hand side of the equation is an increasing function of $q$, enabling us to obtain the value of $q$ that satisfies the equation by binary search. Let $q^*$ denote the (unconstrained) value of $q$ found in this way. Then, to enforce the constraints above, we observe that $q^* > 1$ if and only if $\sum_{i \in S} I_i > \sum_{i \in S} \hat{I}_i$, and similarly that $q^* < 1$ if and only if $\sum_{i \in S} I_i < \sum_{i \in S} \hat{I}_i$. Thus we can write:

$$
F(S) = \begin{cases}
\sum_{i \in S}(I_i \log(q^*) - \log(q^*\hat{I}_i - \hat{I}_i + 1)) & \text{if } \sum_{i \in S} I_i < \sum_{i \in S} \hat{I}_i \\
& \qquad \text{and we are searching for over-estimation bias;} \\
\sum_{i \in S}(I_i \log(q^*) - \log(q^*\hat{I}_i - \hat{I}_i + 1)) & \text{if } \sum_{i \in S} I_i > \sum_{i \in S} \hat{I}_i \\
& \qquad \text{and we are searching for under-estimation bias;} \\
0 & \text{otherwise.}
\end{cases}
$$

### A.3 Conditional Bias Scan Framework Parameters

Table 4 contains all the parameter settings needed to run Conditional Bias Scan.

## B Evaluation Appendices for Section 3

### B.1 Adaptations of the Benchmark Methods for Evaluation

Both GerryFair and MultiAccuracy Boost provide implementations of their methods on GitHub (Neel et al., 2019; Kim et al., 2019b). Our goal was to use their provided code with minimal changes as benchmarks in Sections 3 and 4. However, GerryFair and MultiAccuracy Boost do not provide the functionality to indicate whether to audit for bias in the positive direction (under-estimation bias) or bias in the negative direction (over-estimation bias). For example, when auditing for the subgroup, $S^*$, of a protected class, $A$, with the most anomalous deviation in the false positive rate compared to the analogous subgroup in the non-protected class, GerryFair and MultiAccuracy Boost, in their native form, do not allow one to specify if the deviation

Table 4: Table with all parameters needed to run Conditional Bias Scan.

| Parameter | Purpose | Parameter Attribute Values | Sections for Reference |
|---|---|---|---|
| Membership in Protected Class Indicator Variable ($A$) | Binary attribute which defines whether each individual is a member of the protected class. We wish to identify any biases that are present in the classifier's predictions or recommendations that impact the protected class. | | 2 |
| Scan Type | The subcategory of the scan type. | Separation scan for recommendations; Separation scan for predictions; Sufficiency scan for recommendations; Sufficiency scan for predictions | 2.1 |
| Event Variable ($I$) | The event of interest for the scan. The abstracted event variable must be defined as either the outcome, prediction, or recommendation variable. | $Y$; $P$; $P_{bin}$ | 2, 2.1 |
| Conditional Variable ($C$) | The conditional variable for the scan. The abstracted conditional variable must be defined as either the outcome, prediction, or recommendation variable. | $Y$; $P$; $P_{bin}$ | 2, 2.1 |
| Field value ($z$) of Conditional Variable ($C = z$) | For value-conditional scans, this is the value on which we are conditioning the conditional variable ($C$). Defining a field value results in scans that detect different forms of fairness violations. | None; 0; 1 | 2, 2.2, 2.3, A.1 |
| List of Attributes for Forming Subgroups ($X$) | List of attributes to scan over to form subgroups. | | 2, 2.1, A.1 |
| Direction of Bias | Specifies whether we are detecting under-estimation bias (positive direction) or over-estimation bias (negative direction). | Positive; Negative | 2.1, 2.3, A.1 |
| List of Attributes for Estimating $\hat{I}$ ($X$) | List of attributes used for conditioning when producing $\hat{I}$. In this paper we use the same attributes to form subgroups and produce $\hat{I}$. This does not necessarily have to be the case for all applications of CBS. | | 2.2 |
| Subgroup Complexity Penalty | The non-negative integer-valued scalar penalty that is subtracted from the score function for each subgroup depending on the subgroup's total number of included values for each covariate $X^1 \ldots X^m$, not including covariates for which all values are included. | 0+ (default value: 1) | 2.3 |
| Scan Iterations | Specifies the number of iterations to run the fast subset scanning algorithm. | 1+ (default value: 500) | 2.3, A.1 |

The table above lists the parameter, purpose of the parameter, possible values of the parameter, when applicable, and the sections in our paper where this parameter is described in further detail.

should be an increase (or decrease) of the false positive rate for subgroup $S^*$. This lack of functionality makes the results from CBS substantially different than those returned by GerryFair and MultiAccuracy Boost.

For GerryFair's auditor, given the type of error rate to audit (false negative rate or false positive rate), they train four linear regressions using the features ($X$) as dependent variables with the following four sets of labels:

1. Two linear regressions with the zero set as labels.

2. One linear regression with the labels set to a measurement that assigns positive costs for predictions that deviate in the *positive* direction (when the predictions are greater than the observed global error rate), and negative costs otherwise.

3. One linear regression with the labels set to a measurement that assigns positive costs for predictions that deviate in the *negative* direction (when the predictions are less than the observed global error rate), and negative costs otherwise.

They use the predictions from the linear regressions to flag a subset of data where the predictions from the linear regression trained with the zero set labels are greater than the values predicted by the linear regression trained with the costs representing deviations of the predictions from the observed baseline error rate metric of interest as labels. Two linear regressions are used to estimate deviations of the predictions from the observed error rate baseline, and therefore they form two subgroups: (1) a subgroup with rows that are estimated to have predictions that are greater than the baseline for the metric of interest; and (2) a subgroup with rows that are estimated to have predictions that are less than the baseline for the metric of interest. The original GerryFair implementation uses a custom heuristic to decide which subgroup has more significant biases and returns that subgroup accordingly. The subgroup with the rows that are estimated to have predictions that are greater than the metric of interest more closely aligns with the concept of auditing for bias in the positive direction or auditing for under-estimation bias. Since CBS provides the functionality of auditing for biases of a specific direction, we add an option to GerryFair that allows the auditor to parameterize which direction of bias they are interested in, specifically by overriding GerryFair's automated process described above for deciding which subgroup from the two subgroups that represent over- and under-estimation bias has the largest deviation from the baseline, making GerryFair's results more comparable to CBS.

For each simulation, we ran GerryFair two times, once to detect bias in the form of systematic increases in the false positive rate, and once to detect bias in the form of systematic increases in the false negative rate. In each case, we allow GerryFair to use all covariates ($X$) to make the predictions used to form subgroups, including the protected class category. This resulted in two result sets for GerryFair for each simulation. We present the result set in Section 3 that had the highest overall accuracy for most of the simulations, which is the GerryFair setup for detecting increased false positive rate for subgroups of the protected class $A$. GerryFair returns a subgroup that could contain individuals in both the protected class and the non-protected class. To have the accuracy measurements for GerryFair and CBS be comparable, we filter the subgroup returned by GerryFair to only include individuals in the protected class before calculating the subgroup's accuracy.

MultiAccuracy Boost is an iterative algorithm where, on each iteration, it audits for a subgroup with inaccuracies and then corrects that subgroup's predicted log-odds. More specifically, for each iteration:

1. A custom heuristic is calculated for all rows of data, similar to an absolute residual, where larger values represent a larger deviation between the observed labels and predictions.

2. The residuals of all the rows' predictions and observed outcomes are calculated.

3. The full data is split into a training and holdout set.

4. Three partitions of data are created for the training data, hold out data, and the full dataset:

   (a) A partition containing all the rows.
   (b) A partition containing all the rows with predictions greater than 0.50.
   (c) A partition containing all the rows with predictions less than or equal to 0.50.

5. For each of the partitions of data constructed in Step 4:

   (a) A ridge regression classifier (using $\alpha = 1.0$) is trained using the respective partition in the training data, with the covariates $X$ and the sensitive attribute $A$ as features and the custom heuristic calculated in Step 1 as labels.
   (b) The ridge regression classifier is used to make predictions for the respective partition in the holdout data.

(c) If the average of the predictions multiplied by the residuals for the partition set in the hold out data is greater than $10^{-4}$, then the predicted log-odds for the respective partition in the full dataset is shifted by the predictions multiplied by 0.1.

(d) If the predicted log-odds are updated, the iteration terminates and no other partitions of data are evaluated for that iteration.

The steps above are slightly modified for the scenario of a classifier that produces a singular probability of a positive outcome whereas the original MultiAccuracy Boost was designed for was a bivariate outcome vector from an Inception-ResNet-v1 model. To make MultiAccuracy Boost audit for bias in one direction, when calculating whether a partition of the data's predicted log-odds should be updated using the holdout data to remove an inaccuracy, we override the residuals that are negative with 0. In effect, we only consider rows with negative outcomes when deciding which partition of predictions have inaccuracies that need to be corrected on a given iteration. This was the least invasive modification we could make to MultiAccuracy Boost to have it solely consider bias in the positive direction when deciding which subgroup's predicted log-odds to update. When using this slight adaptation, we see an increase in the overall average accuracy for the simulations by approximately 8% for MultiAccuracy Boost compared to a version of MultiAccuracy Boost without the modification intended to account for directional bias.

Since the auditor and correction method are functioning in tandem, we run all iterations of the algorithm and log each subgroup (i.e., partition) that was detected as needing a correction to its predicted log-odds and its associated score calculated in Step 5c. After the algorithm terminates, we find the partition with the highest score and return its associated partition in the full data set. The decision to return the partition with the highest score across all the iterations of MultiAccuracy Boost in the simulations is motivated by the fact that MultiAccuracy Boost's auditor has no theoretical guarantees of detecting the most inaccurate partition on a specific iteration of the algorithm. Similarly to GerryFair, MultiAccuracy Boost detects a subgroup that contains members of the protected class and non-protected class. We filter all the individuals in the returned subgroup to only contain individuals who are part of the protected class before calculating the accuracy of the returned partition.

One distinction between these methods and CBS is that their auditors were intended to be used in conjunction with another process to improve a classifier or predictions. Therefore, their auditors were designed to have the level of detection accuracy necessary to discern which subgroups or partitions of data need to be corrected, either by modifying the classifier or by post-processing their predicted log-odds. Given that both methods suggest that they can be used for auditing purposes, they are appropriate choices as benchmarks for CBS, but it is important to note that CBS was specifically designed to have a high accuracy for bias detection, whereas that was not necessarily an explicit intention of GerryFair or MultiAccuracy Boost.

### B.2 Explanation of the Additive Term ($\epsilon^{true}$) for the True Log-Odds used in the Generative Model for the Semi-Synthetic Data

For the evaluation simulations described in Section 3, when producing the true log-odds that are used to determine the outcomes, $Y$, and predicted values, $P$ or $P_{bin}$, we add a term to each row's true log-odds, $L_i^{true}$, of a value drawn from a Gaussian distribution $\epsilon_i^{true} \sim \mathcal{N}(0, \sigma_{true})$ where $\sigma_{true} = 0.6$. We add this term to the true log-odds to ensure that when the true probabilities ($\text{expit}(L_i^{true})$) for the rows of $S_{bias}$ in the protected class $A$ are injected with $\mu_{suf}$, this results in a violation of the fairness definition for sufficiency.

To illustrate this further, for the remainder of this section we will focus on sufficiency scan for predictions, but our explanation below is applicable for sufficiency scan for recommendations as well. Sufficiency implies that the outcomes $Y$ are conditionally independent of membership in the protected class $A$ given the predictions $P$ and covariates $X$, that is, $Y \perp A \mid (P, X)$. Assume that we have predictions that are independent of the outcome conditional on the covariates, $Y \perp P \mid X$. Since the outcome is independent of the predictions conditional on the covariates, the definition of sufficiency simplifies to $Y \perp A \mid X$. This simplification of sufficiency reduces sufficiency scans to finding the subgroup in the protected class with the largest base rate difference from its corresponding subgroup in the non-protected class regardless of that subgroup's predictions. Therefore, it is not evaluating sufficiency violations because these base rate differences are independent of the predictions. Consequentially, when there is *no* base rate difference between the protected and non-protected

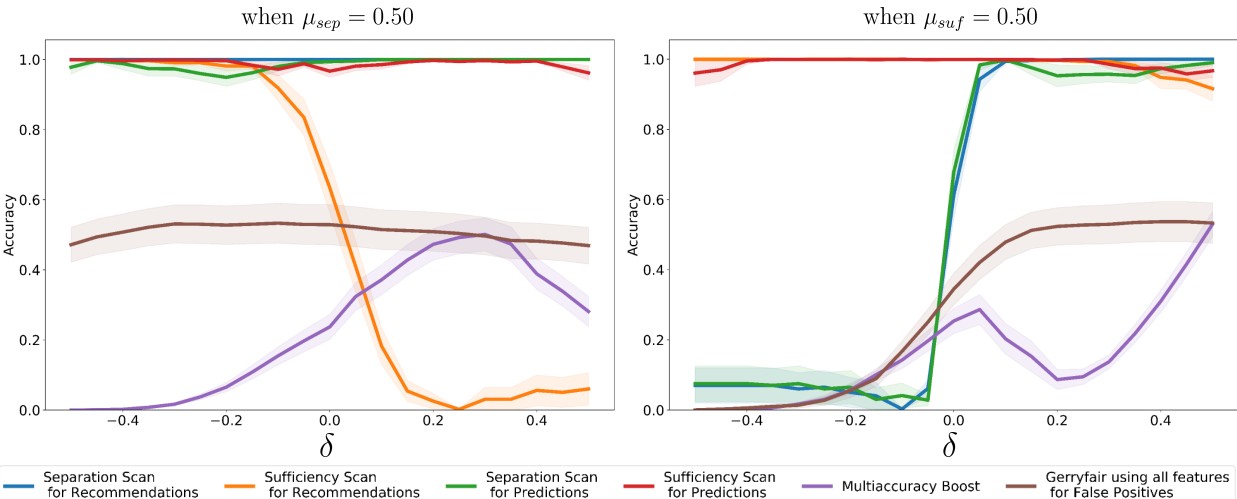

Figure 4: Average accuracy (with 95% CI) for biases injected of $\mu_{sep} = 0.50$ or $\mu_{suf} = 0.50$ into subgroup $S_{bias}$ of the protected class $A$, for CBS, GerryFair, and MultiAccuracy Boost, as a function of varying base rate difference $\delta$ between protected and non-protected class for subgroup $S_{bias}$. Left: increasing predicted probabilities by $\mu_{sep} = 0.50$. Right: decreasing true probabilities by $\mu_{suf} = 0.50$.

class conditional on the covariates, $(Y \perp A \mid X)$, in order for sufficiency to be violated, $Y \not\perp A \mid (P, X)$, we must also have $Y \not\perp P \mid X$. This is formally stated in Theorem B.1.

**Theorem B.1.** *To have violations of the sufficiency definition, $Y \not\perp A \mid (P, X)$, when there are no base rate differences between the protected class and non-protected class conditional on the covariates, $Y \perp A \mid X$, the predictions and outcomes must be conditionally dependent given the covariates, $Y \not\perp P \mid X$.*

*Proof.* Let us assume that (i) there are no base rate differences between protected and non-protected class conditional on the covariates, $Y \perp A \mid X$; (ii) outcomes are independent of the predictions conditional on the covariates, $Y \perp P \mid X$; and (iii) violations of the sufficiency definition exist, $Y \not\perp A \mid (P, X)$. We will show that these three statements lead to a contradiction. First, $(Y \perp P \mid X)$ and $(Y \perp A \mid X)$ together imply that $Y \perp (P, A) \mid X$. Furthermore, using the weak union axiom for conditional independence, $Y \perp (P, A) \mid X$ implies that $Y \perp A \mid (P, X)$, which contradicts (iii). Since these three statements cannot all be true, we know that no base rate differences (i) and violations of sufficiency (iii) together imply that the outcomes cannot be independent of the predictions conditional on the covariates, $Y \not\perp P \mid X$. □

To ensure that $Y \not\perp P \mid X$, the predictions $P$ must carry information about the outcomes $Y$ that is not carried in $X$. By adding the term $\epsilon_i^{true}$ to the true log-odds for each row, given that the predicted log-odds (and the corresponding predicted probabilities $P_i$ and binarized recommendations $P_{i,bin}$) and the outcomes $Y$ are both derived from the true log-odds, this ensures that $Y \not\perp P \mid X$ in the evaluation simulations because $P$ carries information about $Y$, in the form of the added row-wise terms (drawn from a Gaussian distribution), that are captured in $Y$, but are not captured in $X$.

### B.3 Additional Evaluation Simulations

This section includes additional evaluation simulations and their accompanying results conducted to further examine CBS's performance.

Firstly, we investigated the case where we have both an injected bias signal ($\mu_{sep} = 0.50$ or $\mu_{suf} = 0.50$) and an injected base rate shift $\delta$ in subgroup $S_{bias}$ of the protected class $A$ (Figure 4). We examined the extent to which positive and negative shifts for $\delta$ either help or harm the detection accuracy of the various methods.

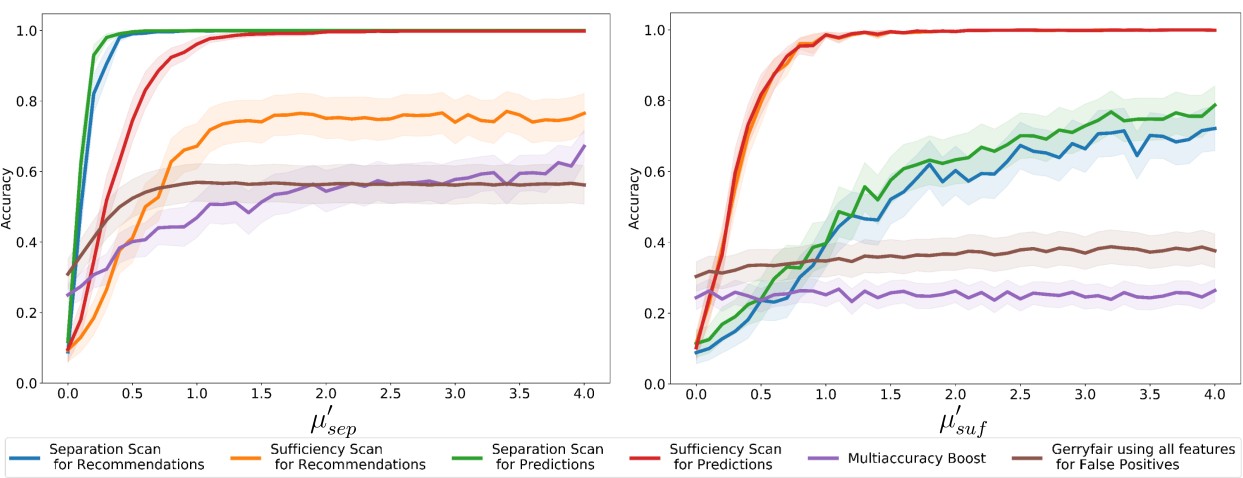

Figure 5: Average accuracy (with 95% CI) as a function of the amount of bias injected into subgroup $S_{bias}$ of the protected class $A$, for four variants of CBS, GerryFair, and MultiAccuracy Boost. Left: increasing predicted log-odds by $\mu'_{sep}$. Right: decreasing true log-odds by $\mu'_{suf}$.

Thus we run two separate sets of experiments, each defined by differing initial setups injecting a fixed and constant bias signal into subgroup $S_{bias}$ for protected class $A$ of: (1) $\mu_{sep} = 0.50$, and (2) $\mu_{suf} = 0.50$. Then, for both experiments, (1) and (2), we vary the injected base rate shift $\delta$ from -0.50 to +0.50. A positive base rate shift ($\delta > 0$) means $S_{bias}$ in the protected class $A$ has a higher base rate, while a negative base rate shift ($\delta < 0$) means $S_{bias}$ in the protected class $A$ has a lower base rate, as compared to $S_{bias}$ in the non-protected class.

In the leftmost plot of Figure 4, where a separation violation is injected with a signal of $\mu_{sep} = 0.50$, we can observe, as expected, that $S_{bias}$ is detected with near-perfect accuracy with separation scans for recommendations and predictions, as well as sufficiency scan for predictions. These results are in line with those displayed in Figure 1, where the injected signal of $\mu_{sep}$ for $S_{bias}$ is increased but the base rate difference is unchanged ($\delta = 0$) for $S_{bias}$ in protected class $A$ compared to $S_{bias}$ in the non-protected class. Additionally, in the leftmost plot of Figure 4, we see that the detection accuracy for the sufficiency scan for recommendations of $S_{bias}$ increases as the base rate difference, $\delta$, for $S_{bias}$ decreases. This is in line with the results contained in Figure 2, where conditioning on the binary recommendation, $P_{i,bin}$, is not sufficient in these simulations to capture the decrease in true probabilities, and, therefore, negative base rate shifts ($\delta \ll 0$) for $S_{bias}$ are detected as sufficiency violations.

In the rightmost plot of Figure 4, where a sufficiency violation is injected with a signal of $\mu_{suf} = 0.50$, the sufficiency scans for recommendations and predictions detect $S_{bias}$ with near-perfect accuracy independent of the base rate shift injected into $S_{bias}$. This is both expected and consistent with the results contained in Figure 1. Additionally, as is documented in prior research and we show empirically in Figure 2, for well-calibrated models, base rate difference in $S_{bias}$ for the protected class $A$ will result in separation violations (Chouldechova, 2017). This is consistent with the results in the rightmost plot of Figure 4 which show that as we inject a positive base rate shift ($\delta \gg 0$) into subgroup $S_{bias}$ of protected class $A$, the separation scans detect $S_{bias}$ with near-perfect accuracy.

Lastly, the method we use in Section 3 for injecting bias signal or shifting the base rate of the affected subgroup $S_{bias}$ in the protected class $A$ involves increasing or decreasing the true probabilities and predicted probabilities. CBS is designed to detect a constant, additive shift in the true and/or predicted log-odds (not the true and/or predicted probabilities) for a subgroup, $S_{bias}$, in the protected class $A$ in comparison to that subgroup in the non-protected class (as shown in the alternative hypotheses contained in Table 2). Therefore, our main simulations are designed to ensure that CBS is robust to injected biases and base rate shifts that do not take the same form as CBS's modeling assumptions, specifically, do **not** take the form

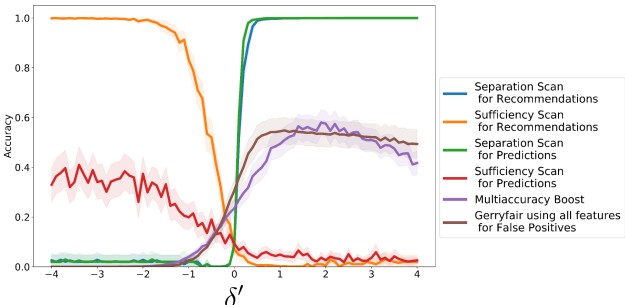

Figure 6: Average accuracy (with 95% CI) as a function of the base rate difference $\delta'$ between protected and non-protected class for subgroup $S_{bias}$, for four variants of CBS, GerryFair, and MultiAccuracy Boost. Note that predictions are well calibrated, $\mu'_{sep} = \mu'_{suf} = 0$.

of additive shifts in the true and/or predicted log-odds. With that said, for comparison purposes, we also examine injected biases and base rate shifts that take the form of shifts in the true and/or predicted log-odds. The resulting Figures 5 and 6 can be directly compared to Figures 1 and 2 respectively. Specifically, we perform the following simulations:

- We increase the predicted log-odds, $L_i^{predict}$, by $\mu'_{sep}$ for $S_{bias}$ in the protected class $A$. Note, this shift is performed prior to the predicted probabilities, $P_i$, being drawn for all the data.

- We decrease the true log-odds, $L_i^{true}$, by $\mu'_{suf}$ for $S_{bias}$ in the protected class $A$. This shift is performed after predicted probabilities have been drawn for all the data. After the true log-odds, $L_i^{true}$, have been decreased by $\mu'_{suf}$ for $S_{bias}$ in the protected class $A$, outcomes $Y_i$ are redrawn specifically for the rows of $S_{bias}$ in the protected class.

- We simultaneously shift the true and predicted log-odds, $L_i^{true}$ and $L_i^{predict}$, by $\delta'$ for $S_{bias}$ in the protected class $A$. Outcomes are redrawn for $S_{bias}$ in the protected class $A$ after the shift by $\delta'$ is performed.

In Figure 5, we observe that the injected signals for $\mu'_{sep}$ and $\mu'_{suf}$ (represented as shifts in the predicted and true log-odds, $L_i^{predict}$ and $L_i^{true}$, respectively) have an effect on CBS's detection accuracy that is nearly identical to the predicted and true probability shifts ($\mu_{sep}$ and $\mu_{suf}$ respectively) shown in Figure 1. Similarly, in Figure 6, we see that the base rate shift created by simultaneously shifting the true and predicted log-odds, $L_i^{true}$ and $L_i^{predict}$, by $\delta'$ for $S_{bias}$ in the protected class $A$ has an effect on CBS's detection accuracy that is nearly identical to the simultaneous shift of the true and predicted probabilities of $S_{bias}$ in the protected class $A$ by $\delta$ as shown in Figure 2. Therefore, we can conclude that CBS performs well for both a constant additive shift in the true and/or predicted log-odds (consistent with its modeling assumptions), as shown in Figures 5 and 6, and also achieves high detection power for non-additive shifts as shown in Section 3, Figures 1 and 2.

### B.4 Robustness Analyses of Evaluation Simulations for Parameters $\sigma_{true}$ and $\sigma_{predict}$

In this section, we examine the robustness of our results in Section 3 by varying the parameters $\sigma_{predict}$ and $\sigma_{true}$ from their default values of 0.2 and 0.6 respectively.

First, we examine the impact of varying $\sigma_{predict}$. Recall that each predicted log-odds, $L_i^{predict}$, is drawn from a Gaussian distribution centered at the true log-odds, with standard deviation $\sigma_{predict}$. Thus $\sigma_{predict}$ can be interpreted as the average amount of random error in the classifier's predictions as compared to the true log-odds values. We run three separate sets of experiments with different initial setups:

(1) We alter $S_{bias}$ in the protected class $A$ by injecting a bias of $\mu_{sep} = 0.50$ (separation violation).

(2) We alter $S_{bias}$ in the protected class $A$ by injecting a bias of $\mu_{suf} = 0.50$ (sufficiency violation).

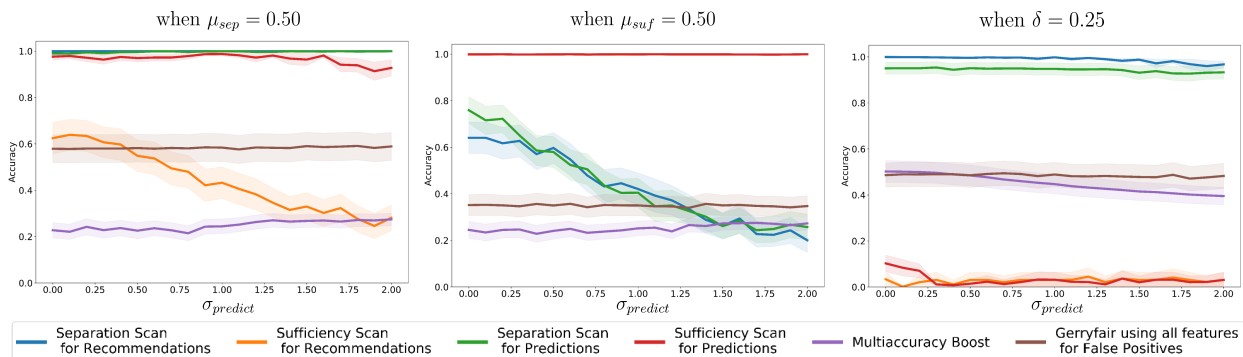

Figure 7: Average accuracy (with 95% CI) for biases and base rate shifts injected into subgroup $S_{bias}$ of the protected class $A$, for CBS, GerryFair, and MultiAccuracy Boost, as a function of varying parameter $\sigma_{predict}$. Left: increasing predicted probabilities by $\mu_{sep} = 0.50$. Center: decreasing true probabilities by $\mu_{suf} = 0.50$. Right: base rate difference $\delta = 0.25$, for $\mu_{sep} = \mu_{suf} = 0$.

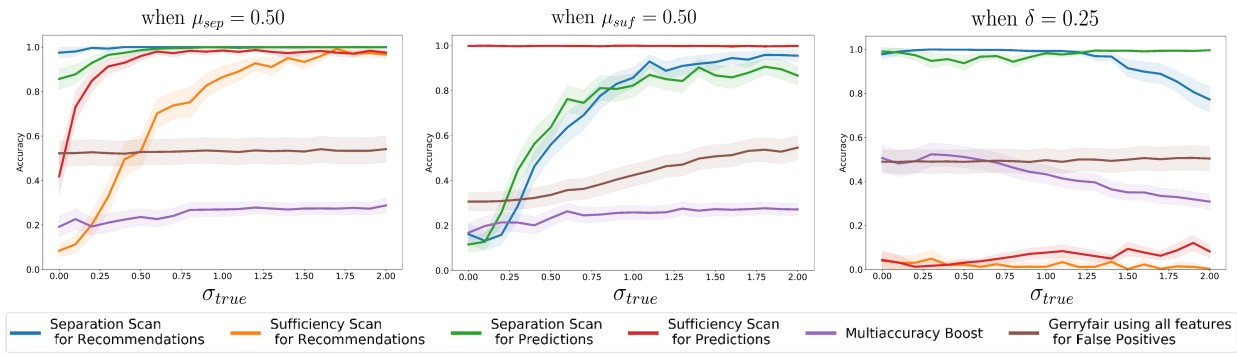

Figure 8: Average accuracy (with 95% CI) for biases and base rate shifts injected into subgroup $S_{bias}$ of the protected class $A$, for CBS, GerryFair, and MultiAccuracy Boost, as a function of varying parameter $\sigma_{true}$. Left: increasing predicted probabilities by $\mu_{sep} = 0.50$. Center: decreasing true probabilities by $\mu_{suf} = 0.50$. Right: base rate difference $\delta = 0.25$, for $\mu_{sep} = \mu_{suf} = 0$.

(3) We create a positive base rate difference for $S_{bias}$ in the protected class $A$ of $\delta = 0.25$.

For each initial setup,((1)-(3)), we run CBS and the competing methods while varying $\sigma_{predict}$ between 0 and 2. Accuracies are averaged over 100 semi-synthetic datasets for each experiment. The experiments where $\mu_{sep} = 0.50$ and $\mu_{suf} = 0.50$ analyze the robustness to $\sigma_{predict}$ of the evaluation simulations for (Q1), whereas the experiments where $\delta = 0.25$ analyze the robustness to $\sigma_{predict}$ of the evaluation simulations for (Q2).

In Figure 7, we observe that large amounts of noise $\sigma_{predict}$ harm the accuracy of the separation scans for injected biases $\mu_{suf} = 0.50$ which shift the true probabilities in subgroup $S_{bias}$ for the protected class. When $\sigma_{predict}$ is large, we see a reduction in accuracy for the sufficiency scan for recommendations for injected biases $\mu_{sep} = 0.50$, which is expected given this scan's initial lower accuracy detection with recommendations with a moderate value of noise in the recommendations.

Second, we examine the impact of varying $\sigma_{true}$. Recall that each individual's true log-odds is a deterministic (linear) function of their covariate values $X_i$ plus a term, $\epsilon_i^{true}$, drawn from a Gaussian distribution centered at 0 with a standard deviation of $\sigma_{true}$. Thus the parameter $\sigma_{true}$ represents the variation between individuals' true log-odds based on characteristics other than the covariate values $X_i$ used by CBS. Moreover, since each individual's predicted log-odds is drawn from a Gaussian distribution centered at the true log-odds, these

characteristics are assumed to be known and incorporated into the classifier, thus creating the dependency $Y \not\perp P \mid X$ when $\sigma_{true} > 0$. In other words, $\sigma_{true}$ represents the average amount of signal in the predictions $P$ (for predicting the outcome $Y$) that is not already present in the covariates $X$.

We run three separate sets of experiments, with the same initial setups described in (1)-(3), where we alter $S_{bias}$ in the protected class $A$ by injecting a bias of $\mu_{sep} = 0.50$, injecting a bias of $\mu_{suf} = 0.50$, and creating a base rate difference of $\delta = 0.25$ respectively. Similarly, for each setup (1)-(3), we run CBS and the competing methods while we vary $\sigma_{true}$ between 0 and 2 for each experiment. Accuracies are averaged over 100 semi-synthetic datasets for each experiment. The experiments where $\mu_{sep} = 0.50$ and $\mu_{suf} = 0.50$ analyze the robustness to $\sigma_{true}$ of the evaluation simulations for (Q1), whereas the experiments where $\delta = 0.25$ analyze the robustness to $\sigma_{true}$ of the evaluation simulations for (Q2).

In Figure 8, we observe that small values of $\sigma_{true}$ harm the accuracy of the separation scans for injected bias $\mu_{suf} = 0.50$ while making them more likely to detect base rate shifts $\delta > 0$ in subgroup $S_{bias}$ for the protected class. Most interestingly, when $\sigma_{true}$ is small, we see a substantial reduction in accuracy for the sufficiency scans for injected bias $\mu_{sep} = 0.50$. This reduced performance for $\sigma_{true} \approx 0$ follows from our argument in Section B.2 above: $\sigma_{true} = 0$ implies $Y \perp P \mid X$, and if we also have no base rate difference between the protected and non-protected classes ($Y \perp A \mid X$), this implies $Y \perp A \mid P, X$. In other words, even if a bias is injected into the predicted probabilities (and recommendations) in subgroup $S_{bias}$ for the protected class, the sufficiency-based definition of fairness is not violated, and thus the injected bias cannot be accurately detected.

## B.5 Estimates of Compute Power

For all of the experiments in Section 3, Appendix B.3, and Appendix B.4, with the exception of the experiments displayed in Figure 5 and Figure 6, we used a university's high-performance computing (HPC) services. We completed all these simulations with 100 jobs that used one node, one core (CPU), and 7 GB of memory each. Each of these jobs performed 1,344 CBS runs, and each job was alive for approximately 9 days. To perform the experiments displayed in Figure 5 and Figure 6, as well as additional robustness checks, we used 15 shared, university compute servers running CentOS with 16-64 cores (CPU) and 16-256 GB of memory. Each server performed 15-120 runs of CBS concurrently, and ran for approximately 9 days. We estimate that to run all of the simulations and robust checks (1,344 CBS runs in total) for a single data set using shifts in the predicted and true probabilities for injecting bias and base rate shifts, this would take approximately 9 days. We estimate that to run all of the simulations and robustness checks (1,504 CBS runs in total) for a single data set using shifts in the predicted and true log-odds for injecting bias and base rate shifts, this would take approximately 32.5 hours. Lastly, to run an individual CBS scan for the COMPAS data (150 iterations), it takes on average approximately 90 seconds. A single run of CBS takes a similar runtime for the German Credit Data.

# C Case Studies Appendices for Section 4

## C.1 Case Study of COMPAS Appendices

### C.1.1 Calibration Curve Plots for Estimating $\hat{I}$ for COMPAS

The method presented in Section 2.2 to estimate the event variable $\hat{I}$ for individuals in the protected class under the null hypothesis $H_0$: $I \perp A \mid (C, X)$ relies on two models being well-specified: (1) the propensity score model for estimating $\Pr(A = 1 \mid X)$; and (2) the outcome model for estimating $\hat{I} = \mathbb{E}[I \mid C, X]$. Figure 9 contains the calibration curves for all propensity score models for all CBS runs on the COMPAS data; note that this curve is computed using all data for fitting $\Pr(A = 1 \mid X)$, which includes all instances in the entire dataset $D$. Figure 10 contains the calibration curves for all outcome models for all CBS runs on the COMPAS data; note that this curve is computed using all data for fitting $\hat{I} = \mathbb{E}[I \mid C, X]$, which only includes the individuals in the non-protected class ($A_i = 0$) and, depending on the scan, may have further restrictions on the conditional variable $C$. For our COMPAS case study, we used logistic regression for all of these models.

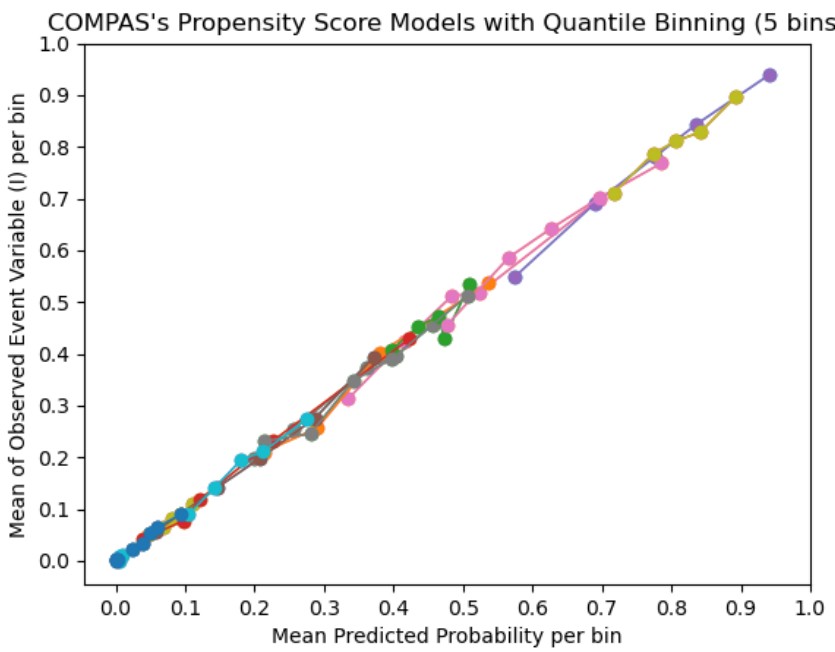

Figure 9: Calibration curves for all propensity score models used to estimate $\Pr(A = 1 \mid X)$ when generating $\hat{I}$, for all CBS runs on the COMPAS data represented in Table 5.

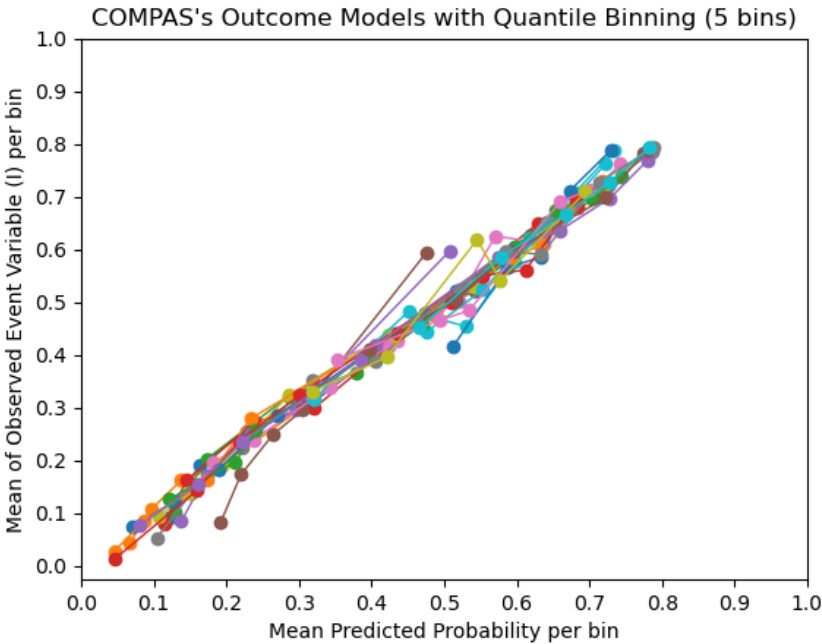

Figure 10: Calibration curves for all outcome models used to estimate $\hat{I} = \mathbb{E}[I \mid C, X]$, using propensity score-weighted data from the non-protected class, for all CBS runs on the COMPAS data represented in Table 5.

### C.1.2 Full Results of COMPAS Case Study

Table 5 contains the full set of COMPAS results for CBS.

| Scan Type | Detected Subgroup ($S^*$) | Log-Likelihood Ratio ($F(S^*)$) | Metric | Observed Metric for Sensitive Detected Subgroup (Num. of Defendants) | Observed Metric for Complement Detected Subgroup (Num. of Defendants) |
|---|---|---|---|---|---|
| \multicolumn{6}{c}{Sensitive Attribute: Black Defendants compared to Non-Black Defendants} |
| Sep. Pred. | Males | **42.4** | FPE | 0.45 (1168) | 0.35 (1433) |
| Sep. Rec. | Males | **102.3** | FPR | 0.44 (1168) | 0.19 (1433) |
| Suff. Pred. | Females | 2.21 | CAL | 0.37 (549) | 0.34 (626) |
| Suff. Rec. | Age 25+ with 0-5 priors | 0.37 | PPV | 0.50 (581) | 0.52 (404) |
| \multicolumn{6}{c}{Sensitive Attribute: White Defendants compared to Non-White Defendants} |
| Sep. Pred. | – | 0.0 | FPE | – | – |
| Sep. Rec. | Females under age 25 with no priors | 2.01 | FPR | 0.71 (31) | 0.56 (70) |
| Suff. Pred. | Under age 25 | 2.36 | CAL | 0.49 (347) | 0.58 (1000) |
| Suff. Rec. | Females under age 25 | 0.41 | PPV | 0.39 (57) | 0.47 (110) |
| \multicolumn{6}{c}{Sensitive Attribute: Native American Defendants compared to Non-Native American Defendants} |
| Sep. Pred. | All Native American defendants | 0.45 | FPE | 0.49 (6) | 0.39 (3357) |
| Sep. Rec. | All Native American defendants | 0.53 | FPR | 0.50 (6) | 0.30 (3357) |
| Suff. Pred. | All Native American defendants | 0.14 | CAL | 0.45 (11) | 0.46 (6161) |
| Suff. Rec. | – | 0.0 | PPV | – | – |
| \multicolumn{6}{c}{Sensitive Attribute: Asian Defendants compared to Non-Asian Defendants} |
| Sep. Pred. | – | 0.0 | FPE | – | – |
| Sep. Rec. | – | 0.0 | FPR | – | – |
| Suff. Pred. | Arrested on misdemeanor charges | **3.16** | CAL | 0.00 (12) | 0.38 (2190) |
| Suff. Rec. | 6+ priors | 0.11 | PPV | 0.00 (1) | 0.76 (965) |
| \multicolumn{6}{c}{Sensitive Attribute: Hispanic Defendants compared to Non-Hispanic Defendants} |
| Sep. Pred. | – | 0.0 | FPE | – | – |
| Sep. Rec. | – | 0.0 | FPR | – | – |
| Suff. Pred. | All Hispanic defendants | 0.26 | CAL | 0.37 (509) | 0.46 (5663) |
| Suff. Rec. | All Hispanic defendants | **2.48** | PPV | 0.56 (141) | 0.63 (2610) |
| \multicolumn{6}{c}{Sensitive Attribute: Male Defendants compared to Female Defendants} |
| Sep. Pred. | Asian | 0.63 | FPE | 0.30 (22) | 0.22 (1) |
| Sep. Rec. | Asian and Hispanic | **22.5** | FPR | 0.21 (286) | 0.05 (57) |
| Suff. Pred. | Native Americans age 25+ | 31.4 | CAL | 0.14 (7) | 1.00 (2) |
| Suff. Rec. | Native Americans age 25+ | 14.1 | PPV | 0.25 (4) | 1.00 (2) |
| \multicolumn{6}{c}{Sensitive Attribute: Female Defendants compared to Male Defendants} |
| Sep. Pred. | White | **1.51** | FPE | 0.38 (312) | 0.35 (969) |
| Sep. Rec. | White | **12.5** | FPR | 0.29 (312) | 0.20 (969) |
| Suff. Pred. | Under age 25 | **18.7** | CAL | 0.38 (246) | 0.60 (1101) |
| Suff. Rec. | Under age 25 | **13.2** | PPV | 0.44 (167) | 0.68 (699) |
| \multicolumn{6}{c}{Sensitive Attribute: Defendants under age 25 compared to Defendants age 25+} |
| Sep. Pred. | All defendants under age 25 | **128.2** | FPE | 0.51 (593) | 0.37 (2770) |
| Sep. Rec. | All defendants under age 25 | **159.3** | FPR | 0.53 (403) | 0.25 (1583) |
| Suff. Pred. | – | 0.0 | CAL | – | – |
| Suff. Rec. | – | 0.0 | PPV | – | – |
| \multicolumn{6}{c}{Sensitive Attribute: Defendants age 25+ compared to Defendants under age 25} |
| Sep. Pred. | – | 0.0 | FPE | – | – |
| Sep. Rec. | Asians arrested on felony charges | 0.74 | FPR | 0.20 (10) | 0.00 (1) |
| Suff. Pred. | Males with 0-5 priors | **92.7** | CAL | 0.35 (2867) | 0.59 (1041) |
| Suff. Rec. | Males with 0-5 priors | **53.0** | PPV | 0.52 (772) | 0.67 (641) |

*Continued on next page...*

| Scan Type | Detected Subgroup ($S^*$) | Log-Likelihood Ratio ($F(S^*)$) | Metric | Observed Metric for Sensitive Detected Subgroup (Num. of Defendants) | Observed Metric for Complement Detected Subgroup (Num. of Defendants) |
|---|---|---|---|---|---|
| Sensitive Attribute: Defendants with no priors compared to Defendants with 1+ priors | | | | | |
| Sep. Pred. | – | 0.0 | FPE | – | – |
| Sep. Rec. | – | 0.0 | FPR | – | – |
| Suff. Pred. | All defendants with no priors | **111.6** | CAL | 0.29 (2085) | 0.54 (4087) |
| Suff. Rec. | All defendants with no priors | **51.0** | PPV | 0.46 (553) | 0.67 (2198) |
| Sensitive Attribute: Defendants with 1-5 priors compared to Defendants with no or 6+ priors | | | | | |
| Sep. Pred. | Under age 25 | **3.28** | FPE | 0.54 (227) | 0.49 (366) |
| Sep. Rec. | Under age 25 | 12.6 | FPR | 0.64 (227) | 0.47 (366) |
| Suff. Pred. | Black defendants of age 25+ | 2.17 | CAL | 0.42 (1038) | 0.55 (1328) |
| Suff. Rec. | Male defendants of age 25+ | **26.8** | PPV | 0.54 (595) | 0.70 (981) |
| Sensitive Attribute: Defendants 6+ priors compared to Defendants with 0-5 priors | | | | | |
| Sep. Pred. | All defendants with 6+ priors | **83.9** | FPE | 0.54 (349) | 0.38 (3014) |
| Sep. Rec. | All defendants with 6+ priors | **126.9** | FPR | 0.66 (349) | 0.26 (3014) |
| Suff. Pred. | – | 0.0 | CAL | – | – |
| Suff. Rec. | Asian | 0.42 | PPV | 0.00 (1) | 0.83 (6) |
| Sensitive Attribute: Defendants arrested on felony charges compared to Defendants arrested on misdemeanor charges | | | | | |
| Sep. Pred. | White females | **2.45** | FPE | 0.42 (139) | 0.34 (173) |
| Sep. Rec. | White females | 9.56 | FPR | 0.38 (139) | 0.21 (173) |
| Suff. Pred. | – | 0.0 | CAL | – | – |
| Suff. Rec. | – | 0.0 | PPV | – | – |
| Sensitive Attribute: Defendants arrested on misdemeanor charges compared to Defendants arrested on felony charges | | | | | |
| Sep. Pred. | – | 0.0 | FPE | – | – |
| Sep. Rec. | Native Americans with 1-5 priors | 1.67 | FPR | 1.00 (2) | 0.00 (1) |
| Suff. Pred. | Females | 3.51 | CAL | 0.26 (491) | 0.41 (684) |
| Suff. Rec. | All defendants arrested on misdemeanor charges | **10.7** | PPV | 0.55 (736) | 0.66 (2015) |

Table 5: Full results from CBS scans run on COMPAS data. Sep. Pred. is short for separation scan for predictions in the positive direction where the metric FPE stands for $\mathbb{E}[P \mid Y = 0, X]$. Sep. Rec. is short for separation scan for recommendations in the positive direction where the metric FPR, i.e. false positive rate, is $\Pr(P_{bin} = 1 \mid Y = 0, X)$. Suff. Pred. is short for sufficiency scan for predictions in the negative direction where the metric CAL, i.e. calibration, is $\Pr(Y = 1 \mid P, X)$. Suff. Rec. is short for sufficiency scan for recommendations in the negative direction where the metric PPV, i.e. positive predictive value, is $\Pr(Y = 1 \mid P_{bin} = 1, X)$. The third column contains the log-likelihood ratio, $F(S^*)$ defined in Equation 1, for the detected subgroup, $S^*$, listed in the second column. Note, bold scores of $F(S^*)$ are statistically significant with p-value $<.05$ measured by permutation testing, as described in Section 2.4. For example, for the separation scan for recommendations with Black defendants as the sensitive attribute (second row), Black males had a false positive rate of 0.44 ($n = 1168$) compared to 0.19 for non-Black males ($n = 1433$).

### C.1.3 Discussion of COMPAS Results for Benchmark Methodologies

Our evaluation of CBS, GerryFair, and MultiAccuracy Boost (Section 3) uses semi-synthetic data that maintains the covariate distribution of COMPAS. The evaluation simulations follow a framework that employs certain generative assumptions for injecting bias into subgroups. The limitations of these generative assumptions used in our framework are discussed in detail in Section 6. In this Appendix, we provide the results of the benchmark methodologies (GerryFair and MultiAccuracy Boost) run on the original COMPAS data, and compare these results to the CBS results for the COMPAS case study in Section 4. We include these results to highlight the differences between CBS and the benchmark methodologies on a non-synthetic dataset, showing the benefits of CBS in a setting without the generative model assumptions used in Section 3.

We ran GerryFair and MultiAccuracy Boost using the same COMPAS data, preprocessing steps, and setup described in Section 4. We report two sets of results: (1) the results of these methodologies (GerryFair and MultiAccuracy Boost) with their out-of-the-box settings; and (2) the results when using the minimum modifications needed to adapt these methods for under-estimation and over-estimation bias, described in Appendix B.1. We include both of these results to display the methodologies' default functionality, which we assume is the intended setting for practitioners, and to obtain a set of results for COMPAS data that can be used to contextualize the differences between these benchmark methodologies and CBS in a real-world setting.

GerryFair and MultiAccuracy Boost provide demonstration code that uses probabilities as the predictive output to be audited, and therefore we use the same $P_i$ calculated for each defendant based on their COMPAS risk score, as described in Section 4.

*GerryFair Results:* When running GerryFair to detect intersectional biases in false positive rates, with race, sex, and the indicator variable of whether defendants are under the age of 25 marked as sensitive attributes, the detected subgroup consists of all defendants aged 25+ who are not Black or Native American. This subgroup is systematically *advantaged* rather than disadvantaged: non-reoffending defendants in the detected subgroup have an average predicted risk $\mathbb{E}(P \,|\, Y = 0) = 0.32$, while non-reoffending defendants not included in this subgroup have an average predicted risk $\mathbb{E}(P \,|\, Y = 0) = 0.45$. When modified to perform a directional scan to search for a systematically disadvantaged subgroup, GerryFair detects a subpopulation consisting of three distinct, marginal groups—all defendants under 25, all Black defendants, and all Native American defendants—rather than an intersectional or contextual subgroup.

*MultiAccuracy Boost Results:* MultiAccuracy Boost chooses between three partitions of data on each iteration of the algorithm, where the chosen partition has its probabilities adjusted. When running MultiAccuracy Boost with its default settings on COMPAS, the highest scoring partition is found on the first iteration. This partition consists of all defendants in the initial iteration that had higher probabilities ($P > 0.50$), and therefore each of those defendants' probabilities gets adjusted depending on MultiAccuracy Boost's custom residual heuristic metric (see Appendix B.1). Given that there are large overlaps in the covariate spaces of the partition that gets its predictions adjusted and the other partitions, the best way to describe this partition's covariate space is based on the coefficients of the classifier used to model the custom residual heuristic metric, as described in Appendix B.1, where larger values contribute to larger adjustments needed to the probabilities of the defendants in the detected subgroup. The factors that are associated with defendants in this partition needing larger adjustments to their probabilities include defendants with no priors and Hispanic defendants. We note that this algorithm is stochastic, but these covariates consistently show a positive association with larger values of the adjustment heuristic.

When running MultiAccuracy Boost using the modifications described in Appendix B.1 to detect directional bias, the highest scoring partition is found on the first iteration of the algorithm. We find that the factors that estimate the level of adjustments needed to the defendant's probabilities include defendants with no priors, Hispanic and Female defendants, defendants of age 25+, and defendants arrested on misdemeanor charges.

*Discussion:* There are several takeaways to highlight about the results of GerryFair and MultiAccuracy Boost for COMPAS:

- GerryFair's original implementation of its auditor does not allow the user to select between detection of over-estimation bias and detection of under-estimation bias. This results in a detected subgroup of non-reoffending defendants that is advantaged rather than disadvantaged, benefiting from lower predicted risk.

- With our modification to detect directional bias, GerryFair finds a large subpopulation consisting of all Black defendants, all Native American defendants, and all defendants under the age of 25. The results of CBS for separation scans for predictions (Appendix C.1.2) show some similarities with GerryFair's results – that is, for each of the three protected classes included in GerryFair's results, the subgroups detected by CBS within the protected class also have positive scores. The major distinction is that GerryFair is *not* detecting intersectional or contextual subgroups within the protected class, such as the subgroup of Black males detected by CBS. In contrast, CBS identifies that non-reoffending Black male defendants have a higher predicted risk compared to non-reoffending non-Black male defendants, and that this identified racial disparity is more significant than the disparity between all non-reoffending Black defendants and all non-reoffending non-Black defendants.

- More generally, GerryFair appears to lack the flexibility of CBS to specify a single protected class and search for intersectional or contextual subgroups within that protected class for whom bias is present. In the given example, it identifies some individuals using characteristics unrelated to race, and the marginal subgroups of all Black defendants who did not reoffend and all Native American defendants

who did not reoffend respectively. This is consistent with our evaluation results in Section 3, in which GerryFair was able to reliably detect marginal biases (for simulation parameter $p_{bias} = 1$) but had low power to detect smaller, more subtle subgroup biases.

- The results of MultiAccuracy Boost suggest that while MultiAccuracy Boost provides a black-box auditor tool, its auditor does not provide interpretable results. This is because the algorithm forms subgroups based only on prediction thresholding, which results in these subgroups having overlapping covariate spaces. This, in combination with the method's inability to audit for specific biases for specified protected class attributes, results in the algorithm neglecting to find important intersectional biases. This is evident from the factors that describe over-estimation bias being defendants of age 25+, defendants with no priors, Hispanic and female defendants, which somewhat aligns with CBS's results for sufficiency scan for predictions for COMPAS, but does not have the capabilities to also find more subtle biases such as the subgroup of Asian defendants arrested on misdemeanor charges affected by over-estimation bias.

In summary, we believe that the above results demonstrate the advantages of CBS as compared to competing methods, as an auditor for detecting intersectional and contextual biases in a real-world context.

## C.2 Case Study of German Credit Data

We present the results of using CBS to audit for predictive bias in algorithmically-generated risk scores for customers in the German Credit Data (Hofmann, 1994). This dataset contains information about 1,000 customers from a German financial institution. Each row of the dataset represents a customer. For each customer, various pieces of demographic, socioeconomic, and financial information are available, as well as a label generated by the financial institution indicating whether each customer is a "good" (trustworthy for credit) or "bad" (untrustworthy for credit) customer. This dataset is often used in the fair machine learning literature to evaluate the predictive bias of models estimating credit risk. This is also the context we assume for these data. We include these appendices to demonstrate the use of CBS for an additional dataset. This case study also provides an example of running CBS on a notably smaller data set: the German Credit Data is less than one sixth of the size of the COMPAS data in terms of rows. Below we provide the same set of results as those shown for COMPAS above.

### C.2.1 Preprocessing of German Credit Data

We use a publicly available version of the German Credit Data that has mapped the keys in the original Statlog data file to their decoded categories (Datahub.io, 2019).

We follow the feature selection and preprocessing methods documented in Kamiran & Calders (2009), which is one of the first publications that used these data for fair machine learning research. For each customer, we use the following information:

- Whether the customer is under age 26 or age 26+.

- Whether the customer owns, rents, or lives in their housing for free.

- The customer's gender and marital status. These were initially coded as one variable. For CBS we create two separate categories for gender and marital status. Additionally, we create two high-level categories for marital status: single or married/separated/divorced/widowed (i.e., "non-single").

- The customer's credit history. We recode this category to the following schema: previously delayed credit/ critical credit/other existing credit or no credit/all credit paid. This involved combining the "no credit/ all credit paid", "all paid", and "existing credit paid" categories because of their overlap. Additionally, we combine previously delayed credit and critical credit/ other existing credit categories because of a lack of clear differences between the categories. The main motivation of these simplifications was to ensure that each category was not overlapping and thus to increase interpretability. We note that there is a lack of granularity specifying if the customer has never

had credit before or has no credit because they have paid off all their previous credit for most of the customers in the data set. This is why we see a correlation between customers being labeled as untrustworthy for credit and customers in the category of "no credit/all paid".

- Whether a customer is considered a trustworthy or untrustworthy customer for credit by the financial institution. An untrustworthy customer is coded as a positive outcome and a trustworthy customer as a negative outcome for consistency with the COMPAS case study's outcome label.

Unlike COMPAS, which provides both an algorithmically-generated risk score and an observed outcome for each row, the German Credit Data only provides the label of whether a customer is trustworthy or untrustworthy for credit, which is commonly used as an outcome variable. To produce the equivalent of an algorithmically-generated risk score for each customer, which we will subsequently audit for predictive bias, we train a logistic regression model using credit history, age (under 26 or age 26+), and housing ownership as predictors and the binary indicator of whether the customer is trustworthy or untrustworthy for credit as the label. We use this model to produce the predicted probability that each customer is untrustworthy for credit. These predicted probabilities, and the corresponding binarized recommendations as to whether each customer is predicted high-risk or low-risk of being untrustworthy for credit, are the predictive risk scores that we audit with CBS. This modeling approach is an example of "fairness through unawareness" because it does not use the two sensitive attributes (gender and marital status) as predictors in training to produce its predictions and recommendations. We will examine whether the predictions and recommendations produced by this model still contain predictive biases, as identified by CBS.

### C.2.2 Scans for the German Credit Data

We preprocessed the outcome variable (whether a customer is trustworthy or untrustworthy for credit) in a similar fashion to the COMPAS outcome variable. A positive outcome represents a less desirable real-world result. For the German Credit Data, this means that a positive outcome represents an observed untrustworthy customer for credit. Therefore, we run the same scans in terms of conditional variables and direction for the German Credit Data that we ran for COMPAS. For the separation scans, we detect positive deviations for the protected class attribute in $\mathbb{E}(P \mid Y = 0, X)$ and $\Pr(P_{bin} = 1 \mid Y = 0, X)$, i.e., increase in average predicted risk for trustworthy customers and increase in FPR (probability of being predicted high-risk for trustworthy customers), respectively. For the sufficiency scans, we detect a negative deviation for the protected class in $\Pr(Y = 1 \mid P, X)$ and $\Pr(Y = 1 \mid P_{bin} = 1, X)$, i.e., decreased probability of being an untrustworthy customer conditional on predicted risk and conditional on being predicted as high-risk, respectively. For the separation and sufficiency scans for recommendations, we threshold the probability risk-scores by 0.5 to construct recommendations: $P_{bin} = \mathbf{1}\{P \geq 0.5\}$. Given the smaller dataset size (as compared to COMPAS) and highly-correlated predictor variables, we found that logistic regression was inadequate for computing propensity scores and for the outcome model (predicting the probabilities $\hat{I}$ using data from the non-protected class). Thus we use a more flexible model– a gradient boosting classifier with Platt scaling – to ensure that our predictions are well-calibrated when computing propensity scores and when estimating $\hat{I}$. All scans were run for 500 iterations with a penalty equal to 1.

### C.2.3 Calibration Curve Plots for Estimating $\hat{I}$ for German Credit Data

The method presented in Section 2.2 to estimate the event variable $\hat{I}$ for individuals in the protected class under the null hypothesis $H_0$: $I \perp A \mid (C, X)$ relies on two models being well-specified: (1) the propensity score model for estimating $\Pr(A = 1 \mid X)$; and (2) the outcome model for estimating $\hat{I} = \mathbb{E}[I \mid C, X]$. Figure 11 contains the calibration curves for all propensity score models for all CBS runs on the German Credit Data; note that this curve is computed using all data for fitting $\Pr(A = 1 \mid X)$, which includes all instances in the entire dataset $D$. Figure 12 contains the calibration curves for all outcome models for all CBS runs on the German Credit Data; note that this curve is computed using all data for fitting $\hat{I} = \mathbb{E}[I \mid C, X]$, which only includes the individuals in the non-protected class $(A_i = 0)$ and, depending on the scan, may have further restrictions on the conditional variable $C$. For our German Credit Data case study, we used gradient boosting classifiers with Platt scaling for all of these models.

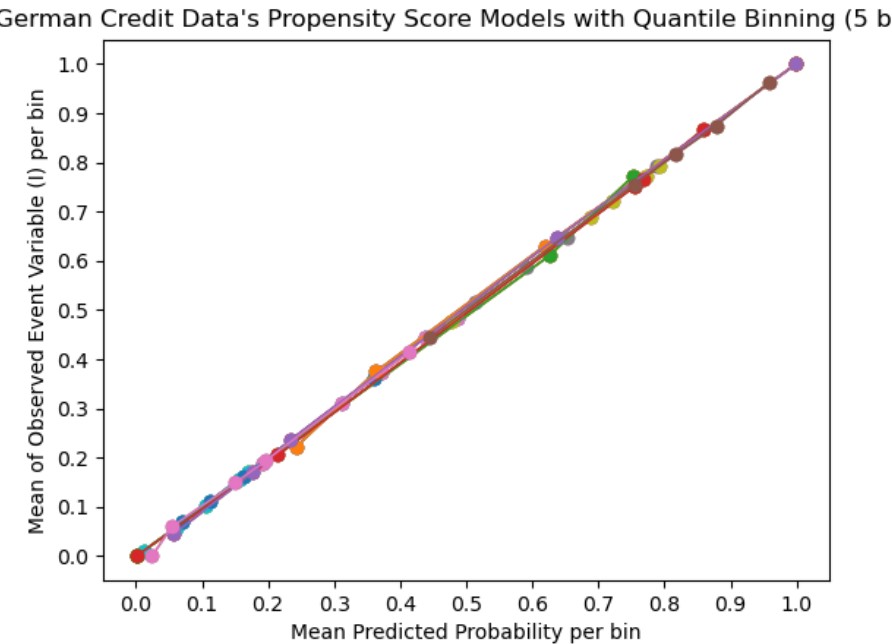

Figure 11: Calibration curves for all propensity score models used to estimate $\Pr(A = 1 \mid X)$ when generating $\hat{I}$, for all CBS runs on the German Credit Data represented in Table 6.

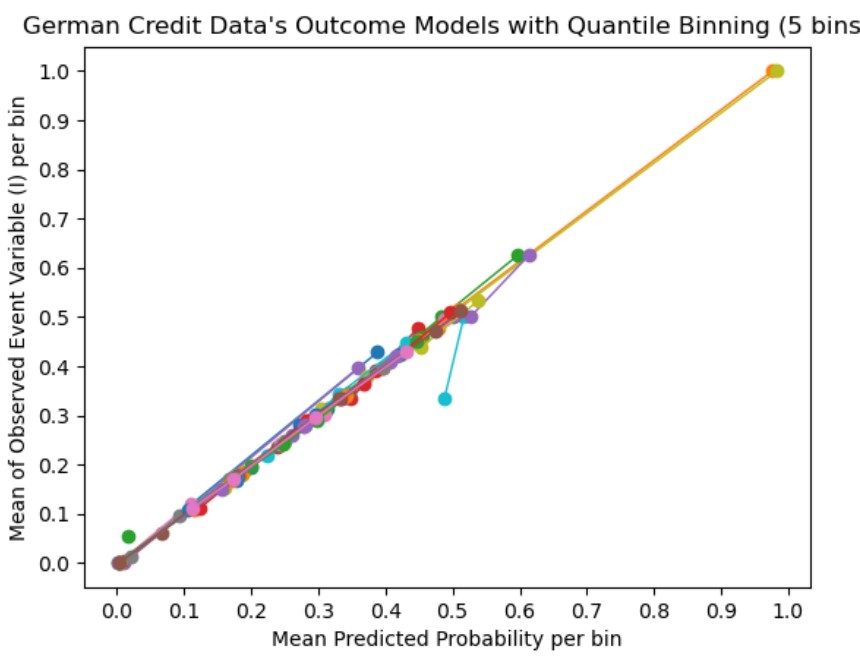

Figure 12: Calibration curves for all outcome models used to estimate $\hat{I} = \mathbb{E}[I \mid C, X]$, using propensity score-weighted data from the non-protected class, for all CBS runs on the German Credit Data represented in Table 6.

### C.2.4   Results of German Credit Data Case Study

Table 6 contains the full set of German Credit Data results for CBS. We observe that the statistically significant biases detected by separation scans are those corresponding to subpopulations with higher base rates (i.e., higher probability of being labeled "untrustworthy" for credit): customers with all paid or no previous credit, younger customers, and customers who have free housing or rent their housing. For sufficiency scans, we detect only a single statistically significant bias: conditional on predicted risk, older female customers with all paid or no previous credit who own their housing are significantly less likely to be labeled as "untrustworthy" than older female customers with all paid or no previous credit who rent or have free housing.

As described in Appendix C.2.1, we purposely excluded the gender and marital status features when modeling the risk scores. Since the exclusion of sensitive features alone does not guarantee that a model will produce predictions without predictive biases, we examine gender biases detected in the logistic regression model's risk scores. It is notable that a sufficiency scan for recommendations identifies a subgroup of female customers who own or rent their housing, have critical, previously delayed, or other existing credit, and are aged 26 or older who are flagged as high-risk for credit. This subgroup has a lower rate of being untrustworthy for credit (0.12) compared to the equivalent group of male customers predicted as high-risk for credit, where the rate of being untrustworthy for credit is 0.19. This scan additionally detects that male customers who have free housing and are predicted as high-risk have a lower rate of being untrustworthy for credit (0.37) as compared to female customers who have free housing and are predicted as high-risk (0.58). Although neither of these detected subgroups is statistically significant, they do represent deviations, in the form of miscalibrated predictions, that disadvantage a subgroup of customers based on their gender as compared to the opposite gender. This suggests that removing gender and marital status as predictors may not be sufficient to fully remove gender-related subgroup biases in the model predictions.

| Scan Type | Detected Subgroup ($S^*$) | Log-Likelihood Ratio ($F(S^*)$) | Metric | Observed Metric for Sensitive Detected Subgroup (Num. of Defendants) | Observed Metric for Complement Detected Subgroup (Num. of Defendants) |
|---|---|---|---|---|---|
| | Sensitive Attribute: Customers under age 26 compared to Customers of age 26+ | | | | |
| Sep. Pred. | All customers under age 26 | **13.5** | FPE | 0.41 (110) | 0.26 (590) |
| Sep. Rec. | – | 0.0 | FPR | – | – |
| Suff. Pred. | All customers under age 26 | 0.07 | CAL | 0.42 (190) | 0.27 (810) |
| Suff. Rec. | – | 0.0 | PPV | – | – |
| | Sensitive Attribute: Customers of age 26+ compared to Customers under age 26 | | | | |
| Sep. Pred. | – | 0.0 | FPE | – | – |
| Sep. Rec. | – | 0.0 | FPR | – | – |
| Suff. Pred. | Single customers who own their housing | 42.4 | CAL | 0.22 (366) | 0.36 (42) |
| Suff. Rec. | – | 0.0 | PPV | – | – |
| | Sensitive Attribute: Female Customers compared to Male Customers | | | | |
| Sep. Pred. | – | 0.0 | FPE | – | – |
| Sep. Rec. | All Female Customers | 0.08 | FPR | 0.11 (201) | 0.03 (499) |
| Suff. Pred. | Owns or rents their housing with critical, previously delayed or other existing credit of age 26+ | 7.31 | CAL | 0.12 (66) | 0.19 (234) |
| Suff. Rec. | – | 0.0 | PPV | – | – |
| | Sensitive Attribute: Male Customers compared to Female Customers | | | | |
| Sep. Pred. | – | 0.0 | FPE | – | – |
| Sep. Rec. | Customers who have free housing under age 26 | 8.39 | FPR | 1.00 (2) | 0.00 (1) |
| Suff. Pred. | Customers who have free housing | 6.23 | CAL | 0.37 (89) | 0.58 (19) |
| Suff. Rec. | All Male customers | 0.02 | PPV | 0.46 (24) | 0.48 (44) |
| | Sensitive Attribute: Single Customers compared to Non-Single Customers | | | | |
| Sep. Pred. | – | 0.0 | FPE | – | – |
| Sep. Rec. | Customers who have free housing under age 26 | 9.19 | FPR | 1.00 (2) | 0.00 (1) |
| Suff. Pred. | Customers who have free housing | 4.92 | CAL | 0.38 (85) | 0.52 (23) |
| Suff. Rec. | – | 0.0 | PPV | – | – |
| | Sensitive Attribute: Non-Single Customers compared to Single Customers | | | | |
| Sep. Pred. | – | 0.0 | FPE | – | – |
| Sep. Rec. | Customers who rent their housing | 2.39 | FPR | 0.42 (74) | 0.09 (35) |
| Suff. Pred. | – | 0.0 | CAL | – | – |
| Suff. Rec. | All Non-Single Customers | 0.54 | PPV | 0.45 (56) | 0.58 (12) |
| | Sensitive Attribute: Customers with all paid or no previous credit compared to Customers with critical, previously delayed or other existing credit | | | | |
| Sep. Pred. | All customers with all paid or no previous credit | **86.5** | FPE | 0.35 (397) | 0.20 (303) |
| Sep. Rec. | – | 0.0 | FPR | – | – |
| Suff. Pred. | Single customers of age 26+ who own their housing | 1.55 | CAL | 0.28 (189) | 0.15 (177) |
| Suff. Rec. | – | 0.0 | PPV | – | – |
| | Sensitive Attribute: Customers with critical, previously delayed or other existing credit compared to Customers with all paid or no previous credit | | | | |
| Sep. Pred. | – | 0.0 | FPE | – | – |
| Sep. Rec. | – | 0.0 | FPR | – | – |
| Suff. Pred. | Customers who own their housing of age 26+ | 8.80 | CAL | 0.16 (267) | 0.29 (340) |
| Suff. Rec. | – | 0.0 | PPV | – | – |

| Scan Type | Detected Subgroup ($S^*$) | Log-Likelihood Ratio ($F(S^*)$) | Metric | Observed Metric for Sensitive Detected Subgroup (Num. of Defendants) | Observed Metric for Complement Detected Subgroup (Num. of Defendants) |
|---|---|---|---|---|---|
| | Sensitive Attribute: Customers with free housing compared to Customers who rent or own their housing | | | | |
| Sep. Pred. | All customers who have free housing | **12.9** | FPE | 0.39 (64) | 0.28 (636) |
| Sep. Rec. | Customers under age 26 | **3.02** | FPR | 0.67 (3) | 0.32 (107) |
| Suff. Pred. | – | 0.0 | CAL | – | – |
| Suff. Rec. | – | 0.0 | PPV | – | – |
| | Sensitive Attribute: Customers who rent compared to Customers with free housing or own their housing | | | | |
| Sep. Pred. | All customers who rent their housing | **5.62** | FPE | 0.38 (109) | 0.27 (591) |
| Sep. Rec. | – | 0.0 | FPR | – | – |
| Suff. Pred. | Female customers | 1.91 | CAL | 0.41 (95) | 0.33 (215) |
| Suff. Rec. | – | 0.0 | PPV | – | – |
| | Sensitive Attribute: Customers who own their housing compared to Customers with free housing or rent their housing | | | | |
| Sep. Pred. | – | 0.0 | FPE | – | – |
| Sep. Rec. | – | 0.0 | FPR | – | – |
| Suff. Pred. | Female customers of age 26+ with all paid or no previous credit | **81.2** | CAL | 0.33 (93) | 0.50 (42) |
| Suff. Rec. | – | 0.0 | PPV | – | – |

Table 6: Full results from CBS scans run on German Credit Data. Sep. Pred. is short for separation scan for predictions in the positive direction where the metric FPE stands for $\mathbb{E}[P \mid Y = 0, X]$. Sep. Rec. is short for separation scan for recommendations in the positive direction where the metric FPR, i.e. false positive rate, is $\Pr(P_{bin} = 1 \mid Y = 0, X)$. Suff. Pred. is short for sufficiency scan for predictions in the negative direction where the metric CAL, i.e. calibration, is $\Pr(Y = 1 \mid P, X)$. Suff. Rec. is short for sufficiency scan for recommendations in the negative direction where the metric PPV, i.e. positive predictive value, is $\Pr(Y = 1 \mid P_{bin} = 1, X)$. The third column contains the log-likelihood ratio, $F(S^*)$ defined in Equation 1, for the detected subgroup, $S^*$, listed in the second column. Note, bold scores of $F(S^*)$ are statistically significant with p-value $<.05$ measured by permutation testing, as described in Section 2.4. Some subgroups are not included for binary sufficiency and binary separation scans because the limited range of the predicted risk score prevented auditing with CBS. These are denoted with a "–" in the Log-Likeliehood Ratio ($F(S^*)$) column. We note that the three lowest-scoring subgroups for sufficiency scan for predictions had higher observed rates in the detected group vs. comparison group. These observed rates were still lower than expected, resulting in small but non-zero scores, given the systematic differences in other predictors between protected and non-protected class. "Non-single" is short for the marital status attribute "Married/divorced/separated/widowed".

### C.2.5 German Credit Data Results for Benchmark Methodologies

We use the same setup described in Appendix C.1.3 for running the benchmark methodologies with their default settings and with the modifications to account for directional bias. Additionally, we use the same data and risk scores described in the other sections of Appendix C.2.

*GerryFair Results:* When running GerryFair with its default settings of detecting positive or negative deviations in the false positive rate in comparison to the global false positive rate with marital status and gender marked as sensitive attributes, GerryFair detects a subgroup of single male customers with a slightly decreased average predicted risk for credit of 0.27 for trustworthy customers in comparison to the global average predicted risk score of 0.29 for trustworthy customers. This is a negative deviation in the false positive rate. The German Credit dataset contains no single females. When running GerryFair to detect positive deviations in the false positive rate, it detects a subgroup of credit-trustworthy married/divorced/separated/widowed customers (i.e., "non-single") who have a slightly increased average predicted risk of 0.30 in comparison to the global expected risk score of 0.29 for all trustworthy customers.

*MultiAccuracy Boost Results:* The MultiAccuracy Boost results, both for its default settings and when accounting for over-estimation bias, found no noteworthy associations between the coefficients of the predictors used to estimate the custom residual heuristic used in MultiAccuracy Boost. This further substantiates our claim that MultiAccuracy Boost does not have the capabilities to be easily used, in terms of interpretability, as an auditing tool for subgroup predictive biases.

