# OpenReview forum: "Auditing Predictive Models for Intersectional Biases"
_TMLR — Accepted by TMLR_

### Review · Reviewer_JpWq · 2025-11-26

**Summary Of Contributions:**

This paper introduces a new framework, the Conditional Bias Scan (CBS), for detecting intersectional and contextual subgroup biases in predictive models. CBS formulates various group fairness notions (separation, sufficiency, prediction-based or recommendation-based fairness) as conditional independence tests and uses fast subset scanning to efficiently search over exponentially many subgroups. The method estimates counterfactual expectations under a null “no-bias” model using propensity reweighting, computes log-likelihood ratio scores for candidate subgroups, and evaluates significance through permutation tests. The paper includes extensive semi-synthetic evaluations, a detailed COMPAS case study, and an additional experiment on the German Credit dataset.

Strengths

* Clear motivation and strong relevance: The paper addresses a central limitation of current fairness audits (attention to only marginal protected attributes) by proposing a principled approach for discovering intersectional biases.

* Methodological novelty: The unification of group fairness criteria into the CBS framework, combined with multidimensional fast subset scanning, is technically original and builds meaningfully on prior work (e.g., Bias Scan).

* Comprehensive empirical evaluation: The semi-synthetic experiments systematically vary bias type, strength, subgroup size, and noise parameters. Detection accuracy comparisons against GerryFair and MultiAccuracy Boost consistently favor CBS.

* Compelling case study: Application to COMPAS reproduces known disparities (e.g., FPR gaps for Black defendants) while revealing more refined intersectional patterns (e.g., Black male non-reoffenders).

* Transparent limitations discussion: The authors appropriately acknowledge assumptions regarding propensity estimation, fairness definitions, and data issues in COMPAS.

Weaknesses / Points for Improvement

* Overstated or imprecise claims. Several statements imply guarantees that exceed what the methods or theory provide: (1) CBS does not reliably identify “the most biased subgroup”; the search is stochastic with only asymptotic optimality guarantees (2) Claims of “significantly higher detection power” are descriptive rather than statistically validated. (3) The framework does not cover “most” group fairness definitions, only those expressible as conditional independence of the form $I\perp A\mid (C,X)$. These should be softened to accurately reflect the scope of guarantees.

* Strong modeling assumptions for estimating $\hat I$. The null expectation $\hat I$ relies on well-specified (and well-calibrated) propensity score and outcome models. Although the paper notes limitations, no diagnostics (e.g., calibration curves, cross-validated predictive performance, sensitivity analyses) are shown for COMPAS. Estimation error in $\hat I$ may materially influence scan results and type I/II error rates.

* Permutation testing approximation. Permutation tests treat the estimated $\hat I$ as fixed across permutations. This neglects uncertainty in the propensity and expectation models; re-estimating them under each permutation would be more principled, though computationally expensive. At minimum, the approximation should be explicitly justified.

* Limited generality of empirical evaluation. The synthetic experiments rely on a single generative family (linear log-odds with Gaussian noise). While varied bias signals are tested, alternative data-generating processes (non-linearities, heteroscedasticity, feature interactions) are not explored. Likewise, only two real datasets are analyzed, both with known limitations.

* Interpretability concerns. Although CBS encourages concise subgroups via a complexity penalty, the paper does not provide guidance on penalty selection or discuss potential instability of subgroup composition under noise or perturbed covariates.

* Missing relevant context. The paper cites intersectionality theory but does not deeply engage with the conceptual gap between sociological intersectionality and statistically defined subgroups. Additional discussion would help prevent overinterpreting statistically detected subgroups as socially meaningful intersectional categories.

Overall Evaluation

This is a high-quality and timely contribution to fairness auditing. The methodological development is rigorous, the empirical work is extensive, and the case studies provide useful insights beyond traditional fairness metrics. The primary weaknesses relate to scope overstatements and unexamined modeling assumptions rather than fundamental flaws. With modest revisions to clarify guarantees, add diagnostic checks for $\hat I$, and temper generality claims, this paper would make a strong contribution to the literature on bias detection and subgroup fairness.

**Audience:**

Yes

**Audience Explanation:**

Yes. At least some of TMLR’s audience would be interested in this paper.
The work addresses an active and important topic in ML (auditing models for subgroup and intersectional bias) which is highly relevant to researchers in fairness, responsible AI, and model evaluation. CBS extends existing subgroup-fairness methods with a more general and efficient framework, and the findings on COMPAS and German Credit provide practical insights that many TMLR readers would care about. Overall, the problem, methodology, and results fall squarely within areas of ongoing interest in the TMLR community.

**Broader Impact Concerns:**

Not sure, sorry

**Claims And Evidence:**

Yes

**Claims Explanation:**

Partially. The core technical claims are supported by solid evidence, but several broader or stronger claims are only weakly supported, e.g.,

* “CBS identifies the subgroup with the most significant bias.”
The experiments illustrate that CBS finds high-scoring subgroups, but the algorithm only guarantees approximate or asymptotic optimality. No experiments compare the discovered subgroups with true global optima on real data.

* “CBS has significantly higher detection power.”
The semi-synthetic evaluation supports comparative performance, but the paper does not perform statistical significance tests across methods or explore a wide diversity of generative models. Results are convincing within the tested scenarios, but generality is overstated.

* “Robust empirical evaluation.”
The simulations are detailed, but robustness claims are limited by:
one generative family (linear log-odds + Gaussian noise),
limited real-world datasets (COMPAS and German Credit),
no stress testing under model misspecification or dependence violations.
Thus the evidence is informative but not “robust” in the broad sense.

*  Claims about “reliability” of subgroup detection.
Reliability depends on accurate estimation of $\hat I$, but the paper presents no calibration or diagnostic analyses for propensity scores or outcome models on COMPAS. The method may underperform if these models are mis-specified.

* Interpretation of sociological “intersectionality.”
The paper meaningfully detects statistical subgroups, but equating these with sociological intersectionality is a conceptual leap. Evidence does not support that CBS captures the structural dynamics described in intersectionality theory.

* Complexity and interpretability claims.
While CBS applies a complexity penalty, the paper does not show experiments on subgroup stability or interpretability trade-offs.

**Requested Changes:**

Clarify and temper overstated claims about optimality and reliability.
Claims such as “CBS identifies the subgroup with the most significant bias” and “reliably detects intersectional biases” should be revised to reflect the stochastic, approximate nature of the scan and its dependence on model specification.

Provide clearer justification and diagnostics for estimating $\hat I$.
Since CBS relies heavily on accurate propensity and outcome models, the paper should include basic diagnostics (e.g., calibration plots or performance metrics) to support their adequacy in the real-data experiments.

Explicitly discuss the limitations of the permutation-testing approximation.
The paper should acknowledge that $\hat I$ is held fixed across permutations and explain the implications for valid $p$-values.

Reframe broad statements about generality and robustness.
General claims (“most fairness definitions,” “robust evaluation”) should be aligned with the actual scope of the experiments and fairness notions covered.

---

> ### Author Response · Authors · 2025-12-22
> **response to reviewer JpWq**
>
> We very much appreciate the attention put into reviewing our manuscript and the detailed feedback provided. Below we outline the various changes we have made to our revised manuscript based on your feedback.
>
> **Clarification on permutation testing.** In our original manuscript, we failed to specify that we *both* re-estimate $\hat{I}$ and run the scan step, after permuting the sensitive attribute $A$, for each simulated dataset during permutation testing.  In our revised manuscript, we have made an edit to Section 2.4 which clarifies that the method for estimating $\hat{I}$, described in Section 2.1, is performed for each simulated dataset during permutation testing. We apologize for this error in the original manuscript.
>
> **Highlighting the gap between an intersectional bias and intersectionality theory.** We agree that the original manuscript lacked precise usage of the terms "intersectional bias" and "intersectionality".  We made edits to the Introduction section to clarify the difference between these two terms and added an additional paragraph to Section 5.4 (within the Related Work section) noting the conceptual gap between an intersectional bias (as used in the algorithmic fairness literature) and the sociological theory of intersectionality.
>
> **Clarifying and refining claims.** We addressed the various issues mentioned in your review pertaining to overstated claims within our original manuscript, including: (1) CBS's ability to reliably find the subgroup of the protected class most affected by bias, given both the stochastic nature of the scan step and the sensitivity our framework has to model misspecification when estimating $\hat{I}$; (2) CBS's detection power being significantly better than the benchmark methods without providing comparative statistical significance testing; (3) claims of a robust evaluation without more varied data generation methods for the semi-synthetic datasets; and (4) claims that CBS is compatible with most fairness definitions and/or group fairness definitions.
>
> We have combed through the original manuscript multiple times, incorporating minor revisions to the language to ensure that either the necessary specificity has been added to our claims to accurately represent CBS's scope, such as clearly stating that our framework is compatible with group fairness definitions that can be represented as conditional independence statements, or rephrasing claims to ensure they are not claiming optimality, reliability, statistical significance or robustness of our evaluation without sufficient supporting evidence.
>
> **Calibration curves for models used to estimate $\hat{I}$.** We understand the importance of the propensity score and outcome models used to estimate $\hat{I}$ being well-specified.  We have added two appendices to our revised manuscript. Appendix C.1.1 contains calibration curves for all of the models (propensity score models in Figure 9 and outcome models in Figure 10) used for estimating $\hat{I}$ in the COMPAS case study, and Appendix C.2.3 contains calibration curves for all of the models (propensity score models in Figure 11 and outcome models in Figure 12) used for estimating $\hat{I}$ in the German Credit Data case study.
>
> Thank you again for these valuable suggestions, which have improved the quality of the paper as well as the clarity and specificity of our writing.

---

### Review · Reviewer_Z978 · 2025-12-03

**Summary Of Contributions:**

The  paper introduces Conditional Bias Scan (CBS), a new auditing framework for detecting intersectional and subgroup biases in predictive models. The experiments conducted on semi-synthetic data and case studies on real COMPAS show that CBS provides higher bias detection power than existing methods (GerryFair, MultiAccuracy Boost) and reveals meaningful intersectional biases

Strengths
- The paper is well-written with clear and meaningful contribution.
- The paper presents an extensive simulation and evaluation (100 semi-synthetic datasets, 1344 experiments) and the proposed auditing framework is compared with too previous frameworks.
- The proposed auditing framework is used on COMPAS dataset (and German credit in the appendix), which reveals interesting intersectional biases (real-world impact).

Weaknesses
- The competitors are old: GerryFair (Kearns et al., 2018) and MultiAccuracy Boost (Kim et al., 2019a).
- “Continuous-valued covariates, X, must be discretized or removed prior to the scan step” (Section 2.2), which may lead to a loss of information.
- The proposed algorithms lack a formal analysis of their computational complexity.

**Audience:**

Yes

**Audience Explanation:**

Machine learning researchers, particularly those working on fairness and responsible AI.

**Broader Impact Concerns:**

There is no ethical concerns.

**Claims And Evidence:**

Yes

**Claims Explanation:**

The experiments conducted on both semi-synthetic data and real-world datasets show the efficiency of the proposed method in comparison with the baseline.

**Requested Changes:**

See the weaknesses.

---

> ### Author Response · Authors · 2025-12-22
> **response to reviewer Z978**
>
> Thank you for reviewing our manuscript and providing valuable feedback!  Below we respond to the weaknesses listed in your review, and where applicable, detail changes to our revised manuscript which account for the requested changes.
>
> **Benchmark methods.**
> Our choice of benchmark methods was constrained to approaches that provide open source implementations, have standalone auditors, detect intersectional biases, and are model-agnostic. GerryFair and MultiAccuracy Boost are canonical methods that satisfy these criteria and have seen sustained adoption in the literature, making them well-established reference points for empirical comparison. We are unaware of other methods with comparable uptake as general-purpose intersectional auditors, which have aligned assumptions and scope with CBS.
>
> **Discretization of covariates $X$ prior to subset scanning.** Within the class of subset scanning and anomalous subgroup detection methods, discretization of covariates is a common modeling assumption used to enable tractable searches over candidate subgroups, and to provide easily explainable results.  For CBS, discretizing the covariates prior to the scan step is essential, as opposed to treating unique continuous variables as their own categorical attribute-value, which could result in overfitting as well as reduced explainability. We have made an edit to the Introduction section in the revised manuscript that highlights the purpose of discretizing covariates prior to the scan step in relation to the explainability of subgroups detected by CBS.
>
> **Analysis of computational complexity.** We have added a new Appendix A.1.3, which includes an analysis of the run time of the Fast Subset Scan step of the CBS framework, and reference this Appendix in the main paper text. Thanks again for these helpful suggestions!

---

### Review · Reviewer_3jt3 · 2025-12-09

**Summary Of Contributions:**

The paper proposes Conditional Bias Scan (CBS), a framework for auditing intersectional biases. The core methodology uses a well-known Fast Subset Scan (FSS) and derives the score function from the log-likelihood test w.r.t. Gaussian and Bernoulli distributions (Neill, 2012). Then the paper provides extensive empirical evaluation and analysis of the framework on the COMPAS dataset, yielding qualitative analysis results that align with prior works, and yielding some new insights.

**Audience:**

Yes

**Audience Explanation:**

Yes. The issue of intersectional bias is of large interest to the ML fairness community.

**Broader Impact Concerns:**

None. The authors have sufficiently addressed this in their Section 6.

**Claims And Evidence:**

Yes

**Claims Explanation:**

Yes.

**Requested Changes:**

**Making the problem setting and writing a bit clearer**
- The definition of subgroup, "a non-empty subset of attribute values for each observed attribute, excluding the sensitive attribute", should be highlighted. Does this imply that the paper only deals with discrete attributes? Because if some attributes are continuous, then each individual would constitute a separate group, right?
- I think this is a misunderstanding on my part, but I'm a bit confused about the setting. So.. there is a single protected attribute $A \in \\{0,1\\}$ and $m$ other covariates, $X^j$ for $j \in [m]$. And the goal is to identify subgroups (e.g., individuals with $X^1 = 1$ and $X^5 = 3$) that are unfairly impacted w.r.t $A$? But then, one of the examples the authors mentioned is "Black and female". My first understanding of this is that there are multiple sensitive attributes, and we are looking for a subset of sensitive attributes (e.g., $A^1$ (ethnicity, black or not black), $A^2$ (gender, female or not female)) that is most impacted by unfairness w.r.t. our chosen criterion (separation, sufficiency).
   - My other possible understanding is that there is indeed only a single binary sensitive attribute $A$, and we are looking for a subset of "binarized" attributes (e.g., $X^1 = 1$ or not; $X^5 = 3$ or not) that "contribute" the most to the unfairness w.r.t. $A$.
   - Anyhow, maybe I missed it, but the current wordy & "flowy" writing style makes it for me too difficult to comprehend what the precise problem formulation is. I am not saying that you should rewrite everything (I personally like such flowy style), but I feel that a bit more rigorous/terse yet-to-the-point problem formulation (sub)section would help a lot.

**Table 2**
- For clarity, although elementary, I think it would help if you could show the derivations of the $F(S)$'s in the Appendix.
- Also, aren't some of them available in closed-form solutions? For instance, with my computations, it seems that $F(S) = \frac{1}{2 \sigma^2 |S|} \left( \sum_{i \in S} \Delta_i \right)^2$ for Gaussian scenario.

---

> ### Author Response · Authors · 2025-12-22
> **response to reviewer 3jt3**
>
> We very much appreciate your feedback on our manuscript, and have addressed all of your requested changes in the revised version. Specific edits are described below.
>
> **Clearer definitions and descriptions of the sensitive attribute and subgroups.**
> The first two requested changes pertain to (1) how CBS handles continuous variables in forming subgroups; (2) the specifics of what a subgroup entails; and (3) the difference between the sensitive attribute ($A$) and the covariates ($X$) used to form subgroups.
>
> Your understanding that there is a single binary sensitive attribute $A$, and that we are looking for a subset of "binarized" attributes (e.g., $X^1 = 1$ or not) that contribute the most to unfairness with respect to $A$, is almost correct, with the distinction that we are looking for a *non-empty subset of values* (e.g., $X^1 \in {1,3,6}$ or not), rather than a single value, for each $X$ attribute.
>
> The following changes to the manuscript address these points. We remain consistent with the original manuscript in excluding notation from the Introduction, but add an example to the Introduction which highlights that there is a single binary sensitive attribute $A$ (female vs. non-female gender, in our example), and that the detected subgroup $S$ consists of subsets of values for other attributes excluding the sensitive attribute. These other attributes can either be sensitive in nature (race/ethnicity and income, in our example)  or non-sensitive. Additionally, this example highlights that continuous variables for the covariates, $X$, must be discretized prior to subgroup scanning (binning income into ranges, in our example), and that a subgroup consists of subsets of attribute values for each covariate excluding the sensitive attribute $A$.
>
> In the overview text for the Methods section (prior to Section 2.1), we sharpen the notation we use by formally defining what a subgroup, $S$, is so that it is clear that a subgroup is defined by the subsets of the covariates' attribute values.  Additionally, we add a note to the end of the overview text for the Methods section highlighting which steps of the framework require the covariates $X$ to be categorical and which steps can be run with categorical and continuous representations for the covariates $X$.  Lastly, we made a small edit to the text to ensure readers understand that the term "sensitive attribute" in the Introduction is referring to $A$ in the Methods section.
>
> **Derivations for score functions $F(S)$.**
> We have added an Appendix A.2 to the revised manuscript that contains the derivations of the Gaussian and Bernoulli score functions $F(S)$. We very much appreciate this valuable suggestion! We note that the score functions are dependent on the constraints contained in Table 2; as shown in Appendix A.2, there is a closed-form for the Gaussian score function $F(S)$ when the maximum likelihood value of the free parameter $\mu$ satisfies the specified constraints.

---

### Author Response · Authors · 2025-12-21
**thank you!**

Thanks very much to the three reviewers for their positive and constructive comments!  We are grateful for your thoughtful feedback and are in the process of incorporating the requested changes into the manuscript, including the addition of calibration curves for $\hat I$, derivations of the Bernoulli and Gaussian score functions, and a brief discussion of the computational complexity of the scan.  We will also make the wording changes suggested by reviewer JpWq, as well as clarifying some minor points.  As requested, we will submit the updated version and response by Tuesday, December 23rd.  Thanks again, and best regards, the Authors.

---

### Decision · Action_Editor_ZFqN · 2026-01-15

**Recommendation:** Accept as is

**Audience:**

Yes

**Audience Explanation:**

The paper proposes useful methodology for auditing intersectional bias, which is an important challenge of fairness-aware machine learning.

**Claims And Evidence:**

Yes

**Claims Explanation:**

The paper introduces Conditional Bias Scan (CBS), a new auditing framework for detecting intersectional and subgroup biases in predictive models. The methodology is sound and is demonstrated in experiments.